# Zika virus prM protein contains cholesterol binding motifs required for virus entry and assembly

Sarah Goellner [1], Giray Enkavi [2], Vibhu Prasad [1], Solène Denolly[1], Sungmin Eu[1,7], Giulia Mizzon [1,3], Leander Witte [1], Waldemar Kulig [2], Zina M. Uckeley[4,8], Teresa M. Lavacca [1], Uta Haselmann[1], Pierre-Yves Lozach[4,9], Britta Brügger [5], Ilpo Vattulainen [2] & Ralf Bartenschlager [1,3,6] ✉

For successful infection of host cells and virion production, enveloped viruses, including Zika virus (ZIKV), extensively rely on cellular lipids. However, how virus protein–lipid interactions contribute to the viral life cycle remains unclear. Here, we employ a chemo-proteomics approach with a bifunctional cholesterol probe and show that cholesterol is closely associated with the ZIKV structural protein prM. Bioinformatic analyses, reverse genetics alongside with photoaffinity labeling assays, and atomistic molecular dynamics simulations identified two functional cholesterol binding motifs within the prM transmembrane domain. Loss of prM–cholesterol association has a bipartite effect reducing ZIKV entry and leading to assembly defects. We propose a model in which membrane-resident M facilitates cholesterol-supported lipid exchange during endosomal entry and, together with cholesterol, creates a platform promoting virion assembly. In summary, we identify a bifunctional role of prM in the ZIKV life cycle by mediating viral entry and virus assembly in a cholesterol-dependent manner.

*Flaviviruses* comprise a large group of enveloped viruses, many being transmitted to a vertebrate host by hematophagous arthropods. Several flaviviruses such as dengue virus (DENV), West Nile virus (WNV), tick-borne encephalitis virus (TBEV), and Zika virus (ZIKV) are medically important human pathogens and represent a global health problem due to the lack of effective treatments along with the high morbidity and mortality caused by infections with these viruses[1].

The flavivirus genome is a positive sense, single-stranded RNA encoding a single open reading frame that is translated into a precursor polyprotein. Processing by viral and cellular proteases creates the three structural proteins [capsid (C), pre-membrane/membrane (prM/M), and envelope (E)] and seven non-structural proteins[2]. While the non-structural proteins are involved in viral genome amplification, assembly, and host immune response evasion[1], the structural proteins together with the viral RNA form infectious virions.

The molecular mechanism(s) involved in ZIKV particle assembly remain largely unknown. Newly synthesized viral RNA is released into the cytosol and together with C, forms the nucleocapsid (NC)[3]. C is released from the polyprotein by sequential cleavage by the host signal peptidase and the viral protease, giving rise to two capsid species: a

[1]Heidelberg University, Medical Faculty Heidelberg, Department of Infectious Diseases, Molecular Virology, Center for Integrative Infectious Diseases Research, Heidelberg, Germany. [2]Department of Physics, University of Helsinki, Helsinki, Finland. [3]German Center for Infection Research (DZIF), Heidelberg partner site, Heidelberg, Germany. [4]Heidelberg University, Medical Faculty Heidelberg, Department of Infectious Diseases, Virology, Center for Integrative Infectious Diseases Research, Heidelberg, Germany. [5]Heidelberg University Biochemistry Center (BZH), Heidelberg, Germany. [6]Division Virus-Associated Carcinogenesis, German Cancer Research Center (DKFZ), Heidelberg, Germany. [7]Present address: d-fine GmbH, Frankfurt, Germany. [8]Present address: Department of Molecular Genetics & Microbiology, University of Florida, Florida, USA. [9]Present address: INRAE, EPHE, IVPC, University of Lyon, Lyon, France. ✉e-mail: ralf.bartenschlager@med.uni-heidelberg.de

membrane-anchored precursor form (amino acids (aa) 1-122) and a fully processed cytosolic form (aa 1-104)[4]. While the precursor associates with lipid droplets (LDs)[5], reported to be decisive for virion assembly[6], the shorter form can get imported into the nucleus altering the cellular transcriptome and ribosome biogenesis[7]. Once the NC has formed, it is enwrapped by an ER-derived lipid envelope decorated with prM and E proteins. The resulting progeny virions are immature, non-infectious, and display a rough outer surface, as prM and E form heterotrimeric spike-like structures. Immature virions translocate through the Golgi, where the low-pH environment triggers irreversible conformational changes, enabling prM cleavage by the cellular protease furin and giving rise to a membrane-resident M protein and a soluble pr fragment. The latter covers the fusion loop in E, preventing premature fusion of progeny virions translocating through the secretory pathway[8]. Once released into the extracellular space, the neutral pH triggers pr peptide release, rendering virus particles infectious. Infection is mediated by E binding to the receptor on the host cell, followed by virion uptake and fusion in late endosomes. Earlier studies have shown that cholesterol incorporated into the DENV envelope mediates viral membrane fusion[9], whereas cholesterol in the endosomal target membrane promotes the fusion activity of WNV[10]. Moreover, interference with de novo synthesis of cholesterol using lipophilic statins significantly reduced the production of infectious ZIKV particles[11]. However, the molecular mechanisms how cholesterol is incorporated into virions and contributes to viral membrane fusion during entry remains unclear.

In this work, we investigate interactions between cholesterol and structural proteins of ZIKV, a prototypic flavivirus. We employ a chemo-proteomic approach that is based on a photo-reactive and clickable cholesterol probe[12] and corroborated obtained results with atomistic molecular dynamics (MD) simulations and reverse genetics. Our results show that the M protein domain of prM has two functional cholesterol-binding motifs having divergent roles in virus entry and assembly. We propose a model in which the M-domain facilitates a cholesterol-supported lipid exchange during endosomal entry and, together with cholesterol creates a platform required for the assembly of infectious virus particles.

## Results

### Efficient cross-linking of cholesterol to the transmembrane domain of the ZIKV prM protein

The critical role of cholesterol in entry and assembly of flaviviruses prompted us to investigate whether the viral structural proteins can interact with cholesterol. To examine this, we used ZIKV as a prototypic flavivirus and employed a bifunctional photoreactive and clickable cholesterol probe (PAC-cholesterol), which has a diazirine at position 6 and a terminal alkyne group on the alkyl side chain (Fig. 1a). While the diazirine group allows crosslinking with proteins in close proximity (<3 Å)[13] via UV irradiation, the alkyne group is used for copper-catalyzed azide-alkyne cycloaddition (CuAAC) of azide-biotin to be used for subsequent affinity purification. For profiling of cholesterol crosslinking to viral proteins, human hepatoma cells (Huh7, subclone Huh7-Lunet[14]) were infected with ZIKV (strain H/PF/2013; MOI = 10) for 24 h, and subsequently treated with the PAC-cholesterol probe. After a 1-h incubation period, cells were UV irradiated and whole cell lysates were subjected to click reaction to add biotin to the cholesterol probe. Biotinylated cholesterol was enriched by pull-down using neutravidin beads and crosslinked co-precipitated proteins were analyzed by western blot (Fig. 1b). While all structural ZIKV proteins were detected in the whole cell lysate, only the prM protein and the transmembrane resident M-domain could be specifically captured along with cross-linked cholesterol (Fig. 1c, d). Although a weak band for E proteins was observed in the eluate, western blot quantifications revealed no significant enrichment when compared to the signal in the no UV control sample (Fig. 1d), indicating only weak association with

the cholesterol probe. As only unspecific bands were detected in the eluate for capsid, quantification was not feasible. Overall, our results argue that amongst the structural proteins only prM and its membrane-resident cleavage product M efficiently associate with cholesterol (Fig. 1d).

To corroborate this result, we investigated incorporation of the cholesterol probe into the viral envelope by using a virus-like particle (VLP) system[15]. Huh7-Lunet T7[16] cells were transfected with a ZIKV prM/E-encoding construct and after 18 h, cells were treated with the PAC-cholesterol probe. After a 4-h incubation period, supernatants were collected and secreted proteins contained therein were concentrated and subjected to click chemistry using an azide fluorescent dye (Fig. 1e). Fluorescent cholesterol cross-linked to proteins was visualized using a fluorescent imager, followed by western blot probing for ZIKV proteins (Fig. 1f). Comparison of culture supernatants from mock and ZIKV prM/E transfected cells identified three fluorescently labeled PAC-cholesterol tagged proteins overlapping with the ZIKV structural proteins, most notably with M, the cleavage product of prM (Fig. 1f).

### Cholesterol binding motifs in the M-domain and their role in cholesterol association

To identify potential cholesterol binding sites in prM, we searched for cholesterol recognition amino acid consensus sequences (CRAC: $[L/V]$-$X_{1\text{-}5}$-$[Y/F]$-$X_{1\text{-}5}$-$[R/K]$) and CARC corresponding to the "inverted" CRAC motif)[17]. We identified one CRAC motif overlapping with one CARC motif within the membrane helix (MH), and two CARC motifs located within the transmembrane segments (TMS) 2 and 3 of the M-domain (Fig. 2a, b). Multiple sequence alignments revealed that the CARC2 motif (amino acids 38-44) at the junction of TMS1 and TMS2 and the CARC3 motif (amino acids 60-69) in TMS3 are highly conserved among flaviviruses (Fig. 2b). Since the presence of such motifs does not predict the ability of the protein to interact with cholesterol[18], we evaluated their functionality by expressing HA-tagged wild-type (WT) prM or mutants thereof in Huh7-Lunet T7 cells. To this end, we replaced the aromatic residue by either alanine (A) or serine (S) and the basic residue by leucine (L), resulting in a total of six mutants (CRAC/CARC1: R23L + Y25A/S + R31L (for simplicity designated as CRAC/CARC 1-A and 1-S); CARC2: R38L + F42A/S (CARC 2-A and 2-S); CARC3: K60L + Y63A/S (CARC 3-A and 3-S)) (Fig. 2b). For all mutants, correct membrane topology of prM was ascertained by N-terminal addition of the capsid anchor sequence that acts as signal sequence (Fig. 2c). To confirm that the HA tag does not affect prM subcellular localization, immunofluorescence microscopy was performed. Huh7-Lunet T7 cells were transfected with individual prM-HA constructs and 18 h later, prM and the ER-resident protein reticulon 3 (RTN3) were detected (Supplementary Fig. 1a). As positive control, ZIKV-infected cells were analyzed in parallel. Our result shows that prM-HA was distributed throughout the cytoplasm and colocalized with RTN3 to a similar extent as observed in infected cells (Supplementary Fig. 1b). Furthermore, to confirm that an HA-tagged prM is functional with respect to ZIKV particle formation, we analyzed whether co-expression of prM-HA and E leads to the secretion of VLPs (Supplementary Fig. 2a, b). Indeed, E and prM-HA were co-secreted from transfected cells, arguing that prM-HA is assembly competent. Consistently, both proteins displayed intense colocalization (Supplementary Fig. 2c, d).

Having confirmed functionality of HA-tagged prM, we next validated the cholesterol binding motifs in the M-domain with respect to their ability of being cross-linked to PAC-cholesterol. To facilitate detection of prM, transfected cells were treated with the proteasome inhibitor MG132 for three hours prior to PAC-cholesterol treatment (Fig. 2c). This was necessary because the expression of prM triggers apoptotic cell death[19], which is counteracted by rapid prM degradation. Under these conditions, PAC-cholesterol pull-down was significantly reduced for each of the two prM CARC2 and CARC3 mutants, whereas for the CRAC/CARC1 mutants crosslinking efficiency was

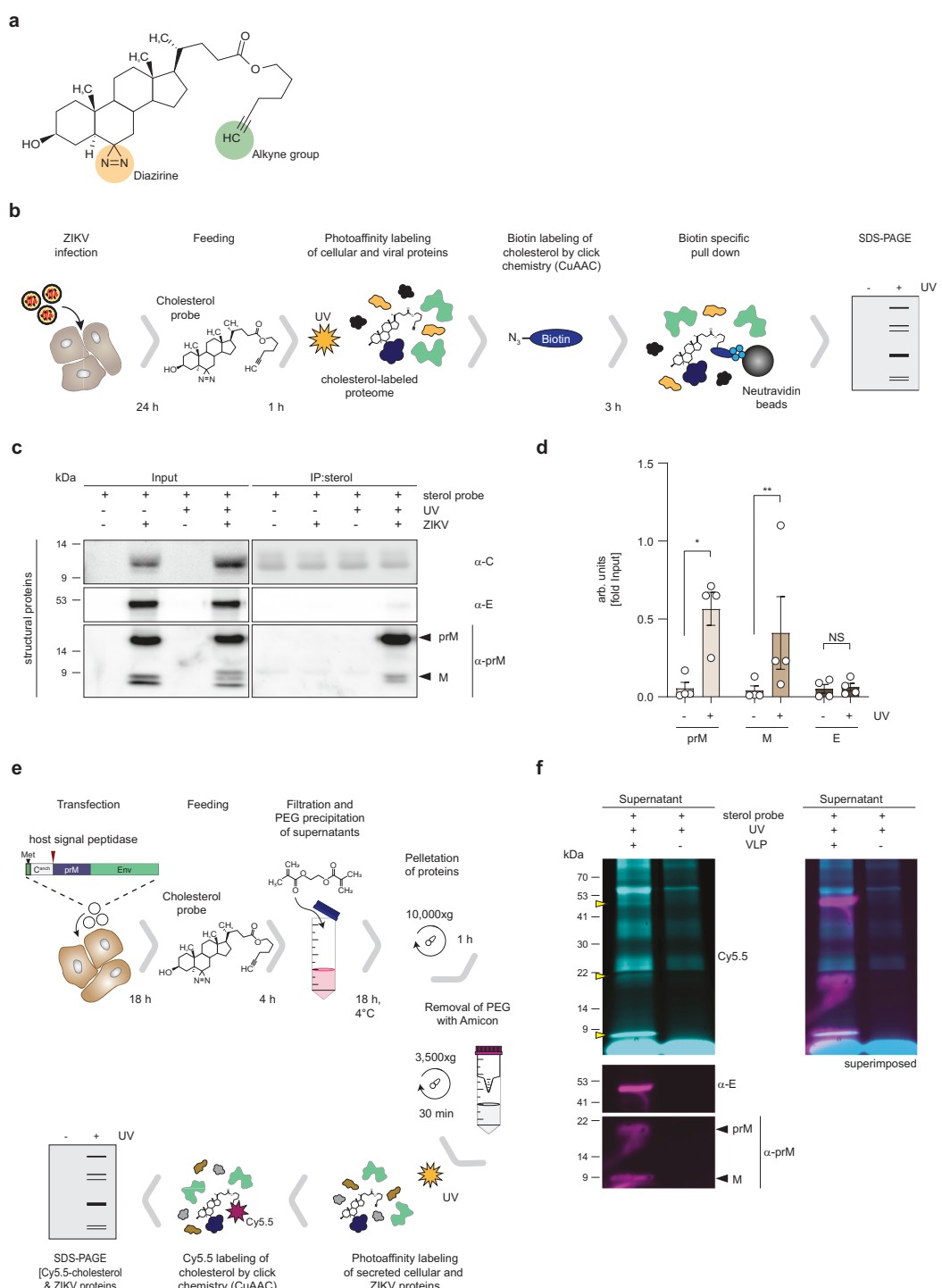

**Fig. 1 | Cholesterol photoaffinity labeling in ZIKV-infected Huh7-Lunet cells identifies prM as likely cholesterol binding protein. a** Chemical structure of the photoreactive and clickable cholesterol probe. **b** Experimental approach. Huh7-Lunet cells were infected (MOI = 10) with ZIKV H/PF/2013. At 24 h.p.i. cells were treated with 10 μM of the PAC-cholesterol probe for one hour. Proteins were crosslinked to the sterol probe by UV irradiation prior to cell lysis and the probe was biotinylated using click chemistry. Protein-sterol-biotin complexes were captured using neutravidin-conjugated resin beads and analyzed by western blot. **c** Detection of ZIKV proteins in captured sterol–protein complexes by western blot using antibodies specified on the right. Molecular weights are indicated on the left (in kilodalton; kDa). A representative result is shown (n = 4). Samples without UV irradiation but otherwise processed as in **b** were used as specificity control. **d** Quantification of signals of co-captured viral proteins from **c**. Data are mean ± SEM of the IP:sterol/input signal ratio, normalized to input. n = 4 independent

experiments. Two-tailed ratio paired t test, *P < 0.05, **P < 0.01. NS not significant. (prM: no UV vs UV, P = 0.0186; M: no UV vs UV, P = 0.0048). **e** Experimental approach. Huh7-Lunet T7 cells were transfected with a ZIKV prM/E VLP construct. At 18 h.p.t. cells were treated with 10 μM of the PAC-cholesterol probe for 4 h. Supernatants were collected and secreted proteins contained therein were precipitated using polyethylenglycol (PEG). On the next day, proteins were pelleted via centrifugation and washed twice with PBS using a 100 K Amicon filtration device. Retained proteins were subjected to UV irradiation to crosslink the sterol probe to proteins and used for click-labeling with an azide fluorescent dye. Protein-sterol-dye complexes were analyzed by western blot. **f** Detection of the cholesterol probe by fluorescent imaging (top left) and ZIKV proteins by western blot using ZIKV-specific antibodies (bottom left). Molecular weights are indicated on the left (in kilodalton; kDa). A representative result is shown (n = 2). Superimposed image (top right) highlights cholesterol-ZIKV proteins signal overlap.

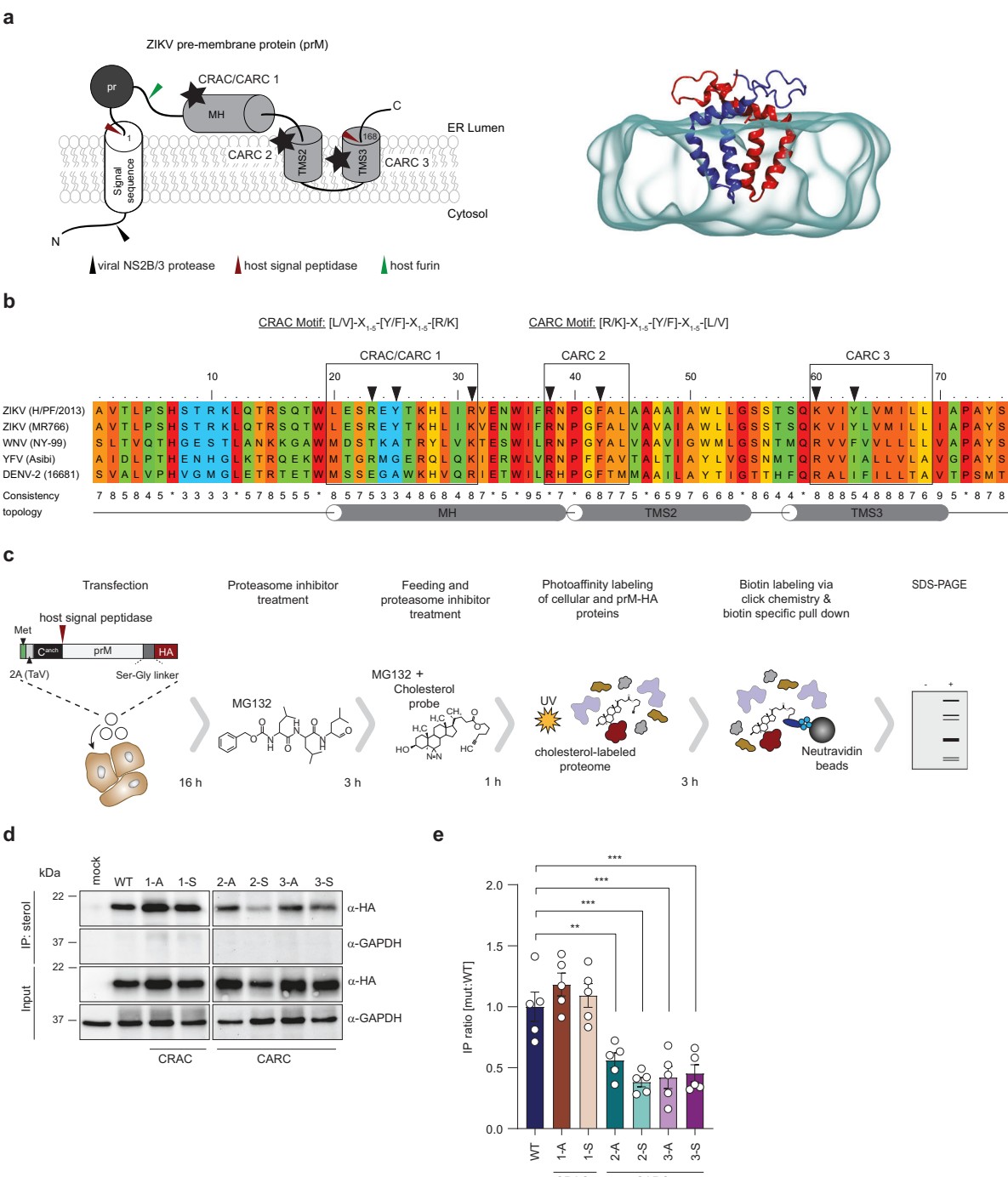

**Fig. 2 | Identification of possible cholesterol-binding motifs in the M-domain of the ZIKV prM protein. a** A schematic representation of the membrane topology of prM is shown on the left. The trans-membrane segments (TMS), the membrane-associated helix (MH) and the signal sequence are represented as gray and white cylinders, respectively. Proteolytic cleavage sites are indicated by arrowheads specified on the bottom. The pr segment is represented as black circle, M is represented in light gray. Stars indicate locations of identified cholesterol binding motifs (CRAC/CARC). A snapshot from the atomistic molecular dynamics simulations of the membrane-embedded M-domain dimer is shown on the right. The monomers of the dimer are shown in blue and red. The hydrophobic core of the lipid membrane is shown in transparent gray. **b** Multiple-sequence alignment of M proteins from different flaviviruses using the PRALINE multiple-sequence alignment software. The residues are colored according to their degree of conservation, ranging from 1 (non-conserved) to 10 (100% conserved; *). Potential cholesterol binding motifs are indicated by boxes and their consensus sequences are given on

the top. Residues selected for mutagenesis are indicated with black arrowheads. **c** Experimental approach. Huh7-Lunet T7 cells were transfected with pTM prM-HA constructs for 16 h. Cells were treated for three hours with 5 µM proteasome inhibitor (MG132), followed by a 1-h incubation with 10 µM PAC-cholesterol probe alongside with 5 µM MG132. After crosslinking by UV irradiation, clarified cell lysates were subjected to biotinylation via click chemistry. Lipid–protein complexes were captured using neutravidin-conjugated resin beads and analyzed by western blot. **d** Representative western blot ($n = 5$). GAPDH was used as loading control. Molecular weights are indicated on the left (in kilodalton; kDa). **e** Relative sterol binding of prM proteins was calculated by densitometry of the HA signals that were normalized to input signals. Data are mean ± SEM of the IP:sterol/input signal ratio, normalized to WT. $n = 5$ independent experiments. One-way ANOVA with Dunnett's test, **$P < 0.01$, ***$P < 0.001$. NS not significant. (WT vs 2-A, $P = 0.0058$; WT vs 2-S, $P = 0.0001$; WT vs 3-A, $P = 0.0003$; WT vs 3-S, $P = 0.0006$).

comparable to the WT (Fig. 2d, e). These results argued for two putative cholesterol binding sites within the M-domain of the prM protein of ZIKV. Since prM and E form a stable complex with E having a possible effect on cholesterol association with M, we repeated the analysis by using the prM/E expression construct supporting VLP formation. Huh7-Lunet T7 cells were transfected, 18 h later treated with MG132 and PAC-cholesterol for 1 h and after cross-linking, PAC-cholesterol was pulled down. Co-precipitated proteins were analyzed by western blot (Supplementary Fig. 3a). While E and prM were equally abundant in whole cell lysate and eluates, cholesterol association of the M proteins in the eluates was reduced with the CARC2 and CARC3 mutants (Supplementary Fig. 3b), thus corroborating the prM-HA results (Fig. 2d, e).

### Atomistic molecular dynamics simulations reveal cholesterol binding to the M-domain

To characterize the cholesterol binding sites and the role of the CARC domains in molecular detail, we performed atomistic molecular dynamics (MD) simulations of the membrane-embedded WT M-domain and the CARC2 and CARC3 mutants. For the simulations, we used previously published cryo-EM data (PDB ID:5IRE) of ZIKV and TBEV virions showing two E-M heterodimers (Fig. 3a), in which the inner core consists of a M protein dimer[20,21], which we extracted and used for subsequent atomistic MD simulations (Fig. 3b). Bioinformatically identified cholesterol-binding motifs were mapped on the extracted M protein dimer, showing that the CARC2 motif is located at the junction of the membrane helix and TMS2 and thus in proximity to the lipid membrane surface, whereas the CARC3 motif is located in the TMS3, embedded in the hydrophobic core of lipid membranes (Fig. 3c). We studied 4 different binary membrane compositions for each protein form: 0:100, 10:90, 20:80, and 30:70 (mol%:mol%) CHOL (cholesterol):POPC (1-palmitoyl-2-oleoyl-phosphatidylcholine) bilayers. In all cholesterol-containing membranes and protein forms, distribution of cholesterol was enhanced next to the surface of the M protein as measured by the radial distribution function (Supplementary Fig. 4a) indicating a favorable interaction of M proteins with cholesterol. Additionally, the average three-dimensional occupancy of cholesterol showed localized density on the surface of the WT M protein dimer (Fig. 3d) corresponding to regions around CARC2 and CARC3 motifs. Interestingly, while the cholesterol binding to WT M protein dimer is clearly seen especially in the 20:80 CHOL:POPC bilayer, the mutants consistently exhibited no specific cholesterol binding in this membrane (Fig. 3e). To characterize distinct cholesterol binding sites, we next clustered protein residues based on their interaction with the same cholesterol. This analysis assigns the protein residues into non-overlapping groups, based on their propensity to interact with the same cholesterol molecules simultaneously, which we refer to as a cholesterol-binding site. This analysis resulted in three distinct binding sites of which site 1 is located within the membrane helix, site 2 in TMS2, and site 3 in TMS3. We next estimated how much each site is occupied by cholesterol, showing a mean occupancy of 0.38 for cluster 2 and 0.35 for cluster 3 in the WT M protein (Supplementary Fig. 4b), whereas site 1 did not substantially interact with cholesterol, consistent with the PAC-cholesterol data. Notably, analysis of the CARC2 and CARC3 mutations revealed a decrease in cholesterol occupancy for sites 2 and 3. Additionally, in 20:80 CHOL:POPC bilayers, surface cholesterol saturation of the M protein dimer was on average about 3–5% (±2%) larger in WT than in the mutants (Fig. 3f). Interestingly, the differences between WT and the mutants were reduced at higher or lower cholesterol concentrations, and some mutants exhibited slightly higher cholesterol coverage than WT (Fig. 3f). To further investigate the effect of mutations in M on cholesterol binding, we constructed cholesterol binding curves based on the surface cholesterol saturations (SCS) analysis (Fig. 3g). The curves clearly indicated stronger binding in WT when compared to mutants.

Here, mutations of CARC2-A and CARC3-A showed the strongest reduction in binding. Taken together, these results suggest that the M-domain interacts with cholesterol. Specificity of this interaction is reduced by the mutations residing in CARC2 and CARC3, but not in the CRAC/CARC1 motif, and affected by the membrane cholesterol levels.

In addition to information about cholesterol association with M, analysis of our simulation data showed that high cholesterol concentrations or mutations in CARC2 and CARC3 did not cause major conformational changes in the M-domain (Supplementary Fig. 4c). Therefore, the cholesterol recognition likely depends on the surface features of the M protein dimer. Nevertheless, further analysis of the simulation systems using Linear Discriminant Analysis (LDA, for details see the Methods section) revealed subtle correlated differences in the conformation of the transmembrane region, which could effectively discriminate between the studied different membrane compositions. The LDA (Supplementary Fig. 4d) captured a narrowing-widening of the transmembrane helices in response to changing cholesterol concentration (Supplementary Movie 1). The accumulation of such small conformational changes in individual M protein dimers might result in a large overall effect in the whole virion.

### Mutations affecting CARC2 and CARC3 in prM reduce ZIKV spread

Having identified two likely cholesterol binding sites within the membrane-resident M-domain of prM, we assessed the role of M–cholesterol interactions in the ZIKV life cycle. To this end, we introduced the CARC2 and CARC3 mutations into the full-length ZIKV genome synZIKV-H/PF/2013 (Fig. 4a) and the corresponding reporter genome synZIKV-R2A-H/PF/2013[22] (Supplementary Fig. 5a). To prevent those mutants from reverting back to WT, amino acid substitutions were designed to comprise three nucleotide substitutions. After transfection of in vitro transcripts corresponding to the viral genomes into Huh7-Lunet cells, viral replication was assessed by either qRT-PCR or by monitoring *Renilla* luciferase activity in 24-h intervals over a period of 120 h. As positive and negative control, WT reporter virus and a replication-deficient mutant (GAA; having an enzymatically inactive RNA-dependent RNA polymerase in the non-structural protein (NS) 5) were included, respectively. Within the first 48 h, no significant differences were detected between CARC mutants and WT ZIKV (Fig. 4b and Supplementary Fig. 5b). Therefore, RNA replication was not affected, which was expected since prM is not involved in this process. However, for later time points when virions released from primary transfected cells started to spread by infecting non-transfected cells in the culture, we observed differences. The most striking difference was the significant reduction of total ZIKV RNA levels as well as *Renilla* luciferase activity with the CARC3 mutants relative to WT (Fig. 4c; Supplementary Fig. 5c). This was not due to different transfection efficiencies (Supplementary Fig. 5d) arguing for impaired virus spread starting around 48 h after transfection. Consistently, immunofluorescence-based visualization of E proteins of transfected Huh7-Lunet cells revealed a significant reduction in the percentage of ZIKV-positive cells for the CARC3 mutants (Fig. 4d, e). In the case of the CARC2 mutants, no significant difference to WT was found when measuring intracellular RNA amounts and the number of ZIKV-positive cells (Fig. 4b-e). However, when we measured the production of infectious virus particles by using TCID50 assay, we observed significantly lower virus titers achieved with the CARC2 mutants relative to WT, but only at 120 h post transfection (h.p.t.) (Supplementary Fig. 6a), which is a time point too late to detect differences in RNA replication and virus spread assays used here (Fig. 4c, e). Consistently, when we generated purified virus stocks of the CARC2 mutants produced in Huh7-Lunet cells 120 h.p.t., titers of infectious virus particles as well as viral genome copy numbers, an approximation for the number of virus particles, were significantly lower than WT (Supplementary Fig. 6b,

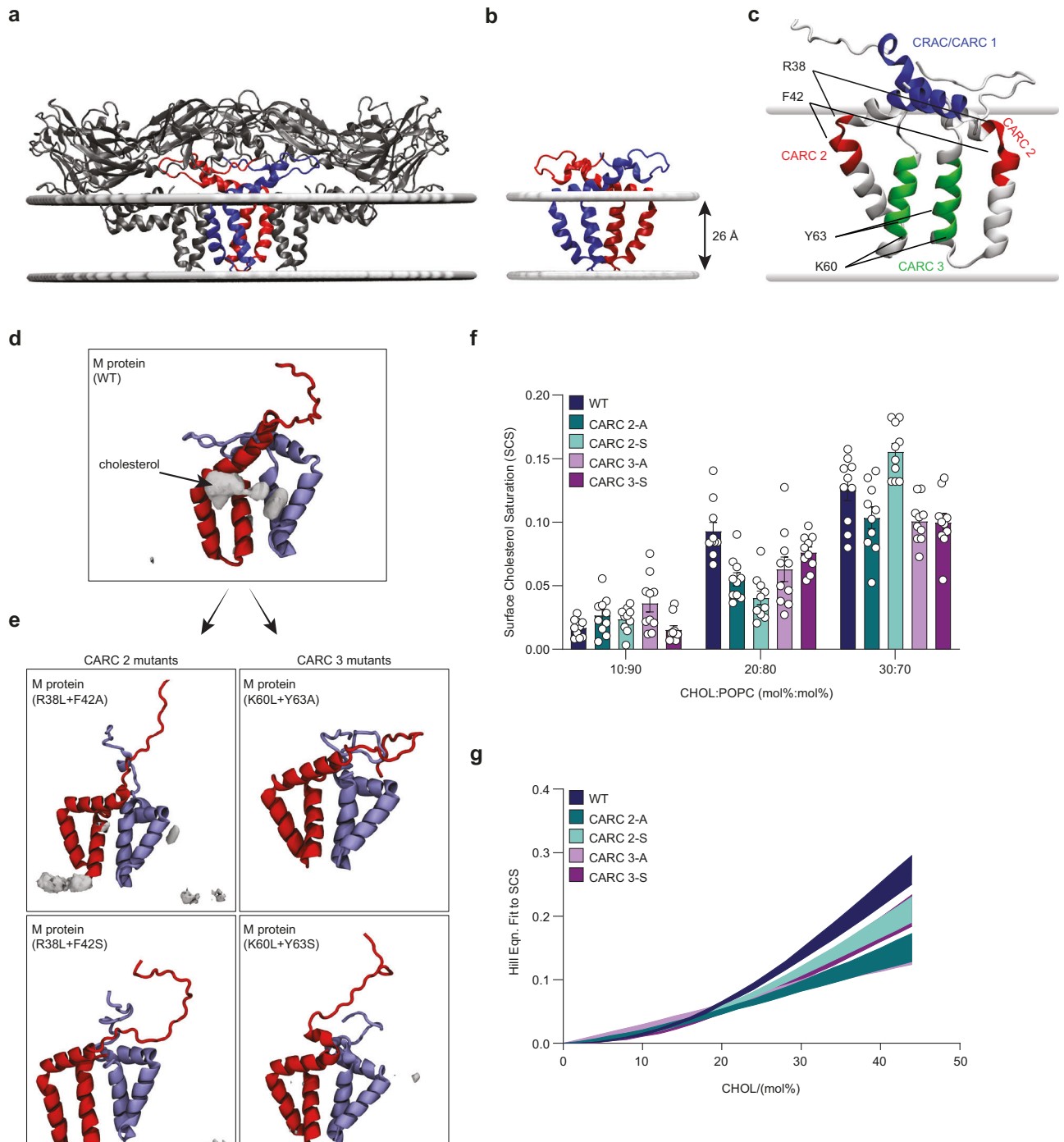

**Fig. 3 | Atomistic molecular dynamics simulations. a** Structural representation of the E-M protein heterodimer in the ZIKV virion embedded into a lipid bilayer (gray lines). The E proteins are depicted in dark gray. The M protein dimer is represented in blue and red. **b** M protein dimer extracted from the heterodimer shown in **a**. This form was used for subsequent atomistic molecular dynamics (MD) simulations. The gray horizontal lines indicate the membrane-embedded hydrophobic part of the protein as calculated using the PPM webserver. **c** The CRAC/CARC motifs identified based on the bioinformatics analysis are indicated on the 3D structure of the M protein dimer embedded into a lipid bilayer (gray lines). **d** MD simulations capture high occupancy of cholesterol (occupancy isosurface at 0.16 is shown) next to WT M protein surface in a 20:80 (mol%:mol%) CHOL:POPC bilayer. The three-dimensional occupancy maps were constructed by averaging data from the last 800 nanoseconds of ten independent repeats, following the removal of translation

and rotation of the protein within the membrane plane. **e** Cholesterol occupancy of given mutations of the CARC2 and CARC3 motifs reveals nearly complete loss of cholesterol binding. **f** Surface Cholesterol Saturation (SCS) is shown as bar plots for each protein variant in membrane compositions specified on the bottom. WT M protein shows significantly more surface saturation in the 20:80 (mol%:mol%) CHOL:POPC system. Averages are taken over ten independent repeats. Error bars indicate SEM. **g** Hill Function fitted to the SCS values obtained for cholesterol-containing systems. WT M-protein responds more strongly to increasing cholesterol concentration than all mutants. Both mutants comprising the alanine substitution (CARC 2-A, 3-A) show the slowest response. The band thickness represents the SEM estimated from 1000 bootstrap samples collected across ten independent repeats.

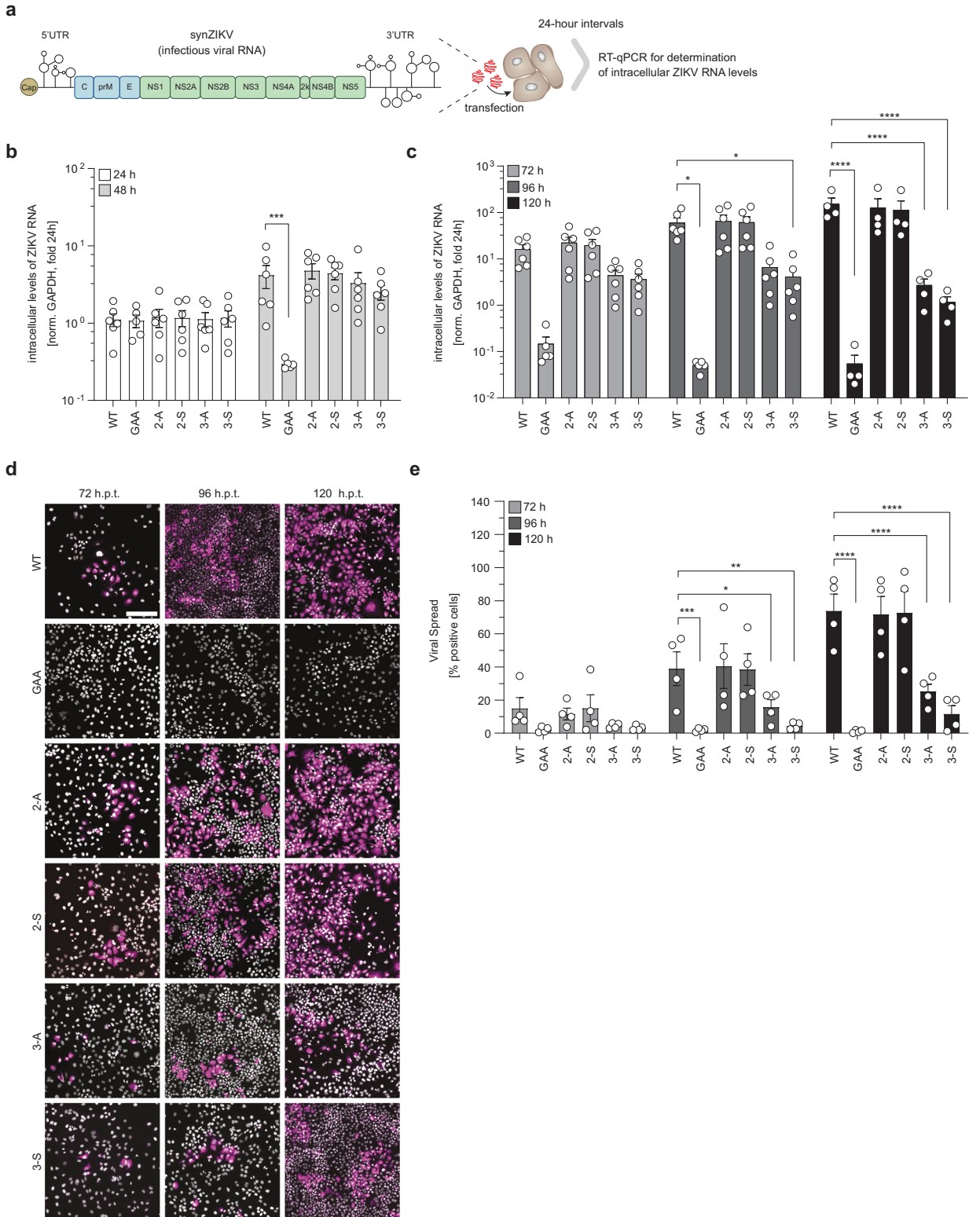

**Fig. 4 | Mutations targeting the cholesterol binding motifs within the M-domain of the prM protein affect virus spread. a** Schematic representation of the synZIKV H/PF/2013 genome. **b**, **c** Huh7-Lunet cells were transfected with in vitro transcripts of synZIKV-H/PF/2013 and intracellular levels of ZIKV RNA were quantified in 24-h intervals over a period of 120 h. Relative intracellular ZIKV RNA levels normalized to GAPDH are plotted. The 24-h values were normalized to their means (**b**). Note that within 48 h very little virus spread occurs and therefore, values reflect RNA replication in transfected cells (**b**). Later time points (72-120 h.p.t.) include virus spread (**c**). Data are mean ± SEM. *n* = 6 independent experiments. Two-way ANOVA with Fisher's LSD test, *$P$ < 0.05, ***$P$ < 0.001,

****$P$ < 0.0001. ((**b**) WT vs GAA, $P$ = 0.0005; (**c**) WT vs GAA, $P$ = 0.0425; WT vs 3-S, $P$ = 0.0474). **d** Viral spread was assessed by immunofluorescence microscopy of cells, fixed 72, 96, and 120 h.p.t. Infected cells were identified by staining the E protein. Representative micrographs are shown (*n* = 4). Scale bar: 200 μm. **e** Quantification of E-positive cells from **d**. Micrographs were analyzed using the FIJI software completed with a self-written macro. Percentages of positive cells are plotted. Data are mean ± SEM. *n* = 4 independent experiments. Two-way ANOVA with Fisher's LSD test, *$P$ < 0.05, **$P$ < 0.01, ***$P$ < 0.001, ****$P$ < 0.0001. (WT vs GAA, $P$ = 0.0007; WT vs 3-A, $P$ = 0.0276; WT vs 3-S, $P$ = 0.0014).

c). Moreover, specific infectivity (ratio of the number of infectious virions to viral RNA genomes) of CARC2 mutant particles was reduced (Supplementary Fig. 6d). These results, together with our MD simulations of the M-domain, suggest that mutations in CARC3 and, to a lesser extent, in CARC2 impair association with cholesterol and reduce ZIKV production and spread.

## Impact of cellular cholesterol levels on infection and replication of ZIKV CARC2 mutants

Our MD simulations implied that cellular cholesterol content affects cholesterol association with M (Fig. 3f). Having used thus far Huh7-Lunet cells that have high cholesterol content, we searched for alternate host cells suitable to address this assumption. Quantifying the total cellular cholesterol content of several mammalian cell lines used for studying ZIKV replication, we found that the placental cell line JEG3 and the lung epithelial cell line A549 had similar total cholesterol amounts as Huh7-Lunet. In contrast, the African green monkey kidney cell line VeroE6 had only ~50% of Huh7-Lunet total cholesterol (Fig. 5a). Therefore, we used VeroE6 cells to investigate the effect of host cholesterol level on the efficiency of ZIKV infection. Virus stocks were produced in Huh7-Lunet cells and used to inoculate VeroE6 cells with ZIKV WT or the CARC2 mutants (MOI = 3). Twenty-four hours later, the number of infected cells was determined by immunofluorescence-based visualization of E proteins (Fig. 5b). Indeed, CARC2 mutants were significantly impaired in their ability to infect VeroE6 cell cultures with a ~99% reduction of positive cells as compared to the WT (Fig. 5c). These results suggest that high cellular cholesterol content compensates the defect of the CARC2 mutants.

The strong attenuation of the CARC2 mutants in low-cholesterol VeroE6 cells might be due to impaired virus entry or inefficient production of infectious virus particles (we did not consider viral RNA replication since prM is not required for this process). Since the first step of viral entry is binding to the host cell, we probed CARC2 mutant particles for binding to VeroE6 cells. To avoid confounding effects that might be caused by different specific infectivities of WT and mutant virus particles, cells were inoculated with the same amount of virus genome equivalents (2,000 ZIKV genome copies per cell). Virus binding was performed on ice for 90 min, followed by extensive washing to remove unbound virus particles, and quantification of viral genome copy numbers associated with VeroE6 cells using the probe-based qRT-PCR approach (Fig. 5d). Virus binding to VeroE6 cells was not affected by the CARC2 mutations, suggesting that some other step of the ZIKV life cycle is impaired.

To exclude that the mutations in the M-domain cause structural changes in E proteins we performed neutralization assays, assuming that structural changes in E would alter accessibility of epitopes to neutralizing antibodies. To this end, we used the E-neutralizing monoclonal antibody 4G2 that recognizes the fusion loop at the extremity of the E ectodomain II (Supplementary Fig. 6e). WT and CARC2 virus particles (5000 genome copies per cell) were incubated for 2 h at 37 °C prior to addition to VeroE6 cells and bound virus particles were quantified by measuring cell-associated viral RNA. CARC2 mutants were equally efficiently neutralized as ZIKV WT (Supplementary Fig. 6e), suggesting that E protein conformation is not profoundly altered by mutations introduced into the M-domain of prM.

To determine whether a step after ZIKV entry is hampered by the CARC2 mutations, we bypassed entry by transfecting VeroE6 cells with the respective reporter virus constructs encoding the *Renilla* luciferase gene. Virus replication was assessed by monitoring *Renilla* luciferase activity in 24-h intervals over a period of 96 h. As positive and negative control, WT and the replication-deficient NS5 mutant (GAA) were included, respectively. Within the first 24 h, when no virus spread occurs and only viral replication in transfected cells is monitored, viral replication was comparable between WT and the CARC2 mutants (Fig. 5e). However, starting with the 48 h' time point, when virus spread

begins, CARC2 signals were lower than WT (Fig. 5e and Supplementary Fig. 6f), suggesting inefficient infection of naïve non-transfected cells.

Previous studies have demonstrated that cholesterol is a crucial factor during flavivirus entry[9,23,24]. Our data suggested that mutations in the CARC2 motif of ZIKV prM affect a step between virion binding and RNA replication in a manner depending on cellular cholesterol level. Since ZIKV enters by receptor-mediated endocytosis, we wondered whether virion escape from the endosome is impaired by CARC2 mutations. To this end, we investigated ZIKV fusion behavior for WT and CARC2 mutants by labeling both virion species with rhodamine (R18)[25], a self-quenching dye that becomes visible only upon membrane fusion (Fig. 5f). To ensure that the R18 labeling does not affect the capability of ZIKV to infect VeroE6 cells, immunofluorescence microscopy assays were performed. To this end, VeroE6 cells were inoculated with ZIKV WT and ZIKV WT-R18 (MOI = 2) and at 24 h.p.i. infected cells were identified by staining double-stranded RNA, an intermediate of viral replication (Supplementary Fig. 7a). Quantification of ZIKV-positive cells revealed that both viruses (WT and WT-R18) infected VeroE6 cells to a similar extent. Given that, R18-labeled ZIKV WT and CARC2-S mutant virions were used to inoculate VeroE6 cells (MOI = 2) and after 2 h cells were fixed and the R18 signal was recorded using fluorescence microscopy. Our results show that in comparison to WT, virions of the CARC2-S mutant fused earlier, as reflected in higher mean signal intensity of CARC2-S fused virions (Fig. 5g, lower left). Consistently, virions of the mutant fused predominantly in proximity to the plasma membrane (PM), whereas WT virions were detected primarily in PM distal sites, as reflected in the sum of distances of all fused virions relative to the cell centroid (Fig. 5g, lower right). We note that the observed R18 signal was specific for virus fusion events, because bafilomycin A1 preventing acidification of endosomes and thus hampering ZIKV fusion[26] reduced the R18 signal close to background (Supplementary Fig. 7b).

Given the finding that lower cholesterol levels in host target membranes triggered premature fusion of the CARC2-S mutant, we wondered whether pharmacological lowering of cholesterol levels in membranes of VeroE6 cells leads to premature fusion of ZIKV WT. For this VeroE6 cells were pretreated for one hour with non-cytotoxic concentrations of the cholesterol solubilizing drug methyl-β-cyclodextrin (MβCD) (Supplementary Fig. 7c), prior to inoculation with WT-R18-labeled virions (MOI = 2). Our results show that extraction of cholesterol from host target membranes leads to premature fusion events in proximity to the PM (Supplementary Fig. 7d). These results prompted us to further investigate whether similar pharmacological lowering of cholesterol levels in the membranes of the high-cholesterol Huh7-Lunet cells would impair infection capability of CARC2 mutants more than ZIKV WT. To address this assumption, Huh7-Lunet cells were treated with non-cytotoxic concentrations of MβCD for 3 h (Supplementary Fig. 7e), prior to infection with the CARC2 mutants or WT ZIKV (MOI = 2). Analysis of cells by immunofluorescence-based detection of E proteins revealed that WT infection was reduced in a dose-dependent manner (Supplementary Fig. 7f), consistent with earlier studies[23]. Notably, CARC2 mutants were much more sensitive to MβCD as revealed by the stronger decrease of ZIKV-positive cells relative to the non-treated control (Supplementary Fig. 7f). Overall, these results support the entry defect of the CARC2 mutants and suggest that in addition to the acidic pH, the cholesterol content of the host target membrane defines when the endosomal membrane and the viral envelope fuse. This conclusion implies that impairing cholesterol−M interaction, as in the case with the CARC2 mutant, might trigger premature, non-productive viral entry.

## No cholesterol dependency of ZIKV CARC2 mutants in mosquito cells

Unlike mammalian cells, insects are incapable of de novo synthesis of cholesterol due to the lack of essential enzymes involved in the

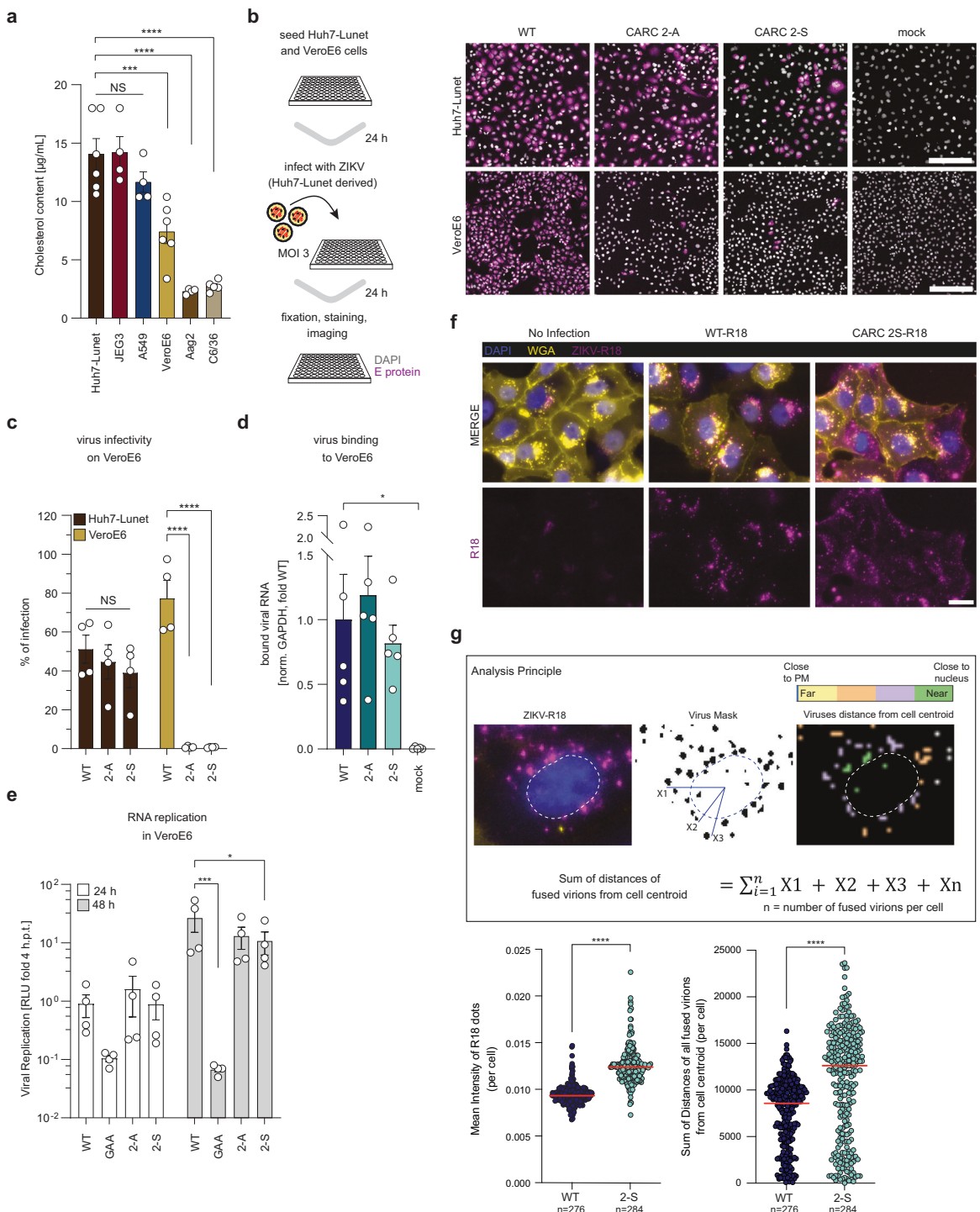

**Fig. 5 | Impact of cellular cholesterol levels on infection competence of CARC2 mutants. a** Quantification of cellular cholesterol content of mammalian and mosquito cell lines. Data are mean ± SEM (*n* = 4-6). One-way ANOVA with Dunnett's test, \*\*\**P* < 0.001, \*\*\*\**P* < 0.0001. NS not significant. (Huh7-Lunet vs VeroE6, *P* = 0.0002). **b** Immunofluorescence microscopy to assess viral infectivity of CARC2 mutants on Huh7-Lunet (positive control) and VeroE6 cells. Cells were infected with WT or CARC2 mutants (MOI = 3) and infection was assessed by detection of the E protein 24 h.p.i. Representative micrographs are shown (*n* = 4). Scale bar: 200 μm. **c** Quantification of positive cells from (b). Data are mean ± SEM (*n* = 4). Two-way ANOVA with Fisher's LSD test, \*\*\*\**P* < 0.0001. NS not significant. **d** Virus binding to VeroE6 cells. Precooled cell monolayers were inoculated with WT or CARC2 mutants using the same genome equivalents (2000 copies/cell). After 90 min total RNA was extracted, and virus RNA contained therein was quantified by probe-based qRT-PCR (*n* = 5). Values were normalized to GAPDH and then to WT. Data are means ± SEM. One-way ANOVA with Dunnett's test, \**P* < 0.05. (WT vs mock,

*P* = 0.0267). **e** Viral replication in VeroE6 cells was monitored after transfection with in vitro transcripts of the ZIKV reporter virus genome synZIKV-R2A by measuring *Renilla* luciferase activity at 24 and 48 h.p.t. Relative light units (RLUs) normalized to the 4-h value reflecting transfection efficiency are blotted (*n* = 4). Data are mean ± SEM. Two-way ANOVA with Fisher's LSD test, \**P* < 0.05, \*\*\**P* < 0.001 (WT vs GAA, *P* = 0.0006; WT vs 3-S, *P* = 0.0278). **f** VeroE6 cells were infected with either R18 labeled ZIKV WT or CARC2 mutant virions (MOI = 2) and visualized by immunofluorescence imaging. DAPI staining indicates nuclei, wheat germ agglutinin (WGA) conjugated with Alexa fluor 488 indicates plasma membrane. Red puncta indicate fusion events of R18-labeled virions. **g** At 2 h.p.i. R18 positive events were analyzed with respect to signal intensity and distance relative to the cell centroid (nucleus) using the Cell Profiler software package (*n* = 4). In total 276 events were analyzed for WT and 284 events for the CARC 2-S mutant. Data are mean ± median. Mann-Whitney two-tailed test, \*\*\*\**P* < 0.0001.

cholesterol biosynthesis pathway[27]. Consistently, total cellular cholesterol in the mosquito cell lines Aag2 and C6/36 was >80% lower as compared to Huh7-Lunet cells (Fig. 5a). Flavivirus infection in mosquito cells was found to occur by both fusion at the PM[28] and receptor-mediated endocytosis[29,30], in contrast to sole receptor-mediated endocytosis in mammalian cells[31]. Comparative analysis of lipidomic profiles of DENV virions grown either in mosquito or mammalian cells, revealed differential incorporation of lipids into the virus envelope[32]. While DENV virions produced in mammalian cells were found to contain phosphatidylserine (PS) as major lipid, phosphatidylethanolamine (PE) was the major lipid found in mosquito-grown DENV virions. Furthermore, the cryo-electron microscopy (EM) study using purified mosquito-derived Spondweni virus particles reconstructed PE in a pocket, which is formed near the TMS of E[33]. Although a comparative analysis of the lipid envelope of ZIKV grown in mosquito and mammalian cells is not available, reconstruction of ZIKV virions derived from mammalian and mosquito cells revealed a unique second lipid pocket that is formed by the TMS of E and M proteins in virions derived from mammalian cells[34]. It was reported that this unique lipid pocket in mammalian cells-grown virions accommodates a crooked lipid, which is present in mammalian, but not mosquito cells. These reports and our data described so far suggest that ZIKV cholesterol dependence differs between ZIKV hosts, the extreme being mosquito cells that are devoid of this lipid. To confirm that ZIKV infection does not depend on cholesterol in mosquito cells, we used the cell line Aag2, derived from the natural vector *Aedes aegypti*, and comparatively analyzed ZIKV growth kinetics in cells cultured either in delipidated or lipidated (regular) FBS (Supplementary Fig. 8a). Cells were infected with ZIKV (MOI = 0.05 to allow extensive virus spread), and intracellular viral RNA levels were quantified in 24-h intervals over a period of 96 h by qRT-PCR (Supplementary Fig. 8a). We observed well-comparable growth curves between ZIKV grown in DL-FBS and FBS. Thus, ZIKV does not rely on cholesterol when replicating in mosquito cells, suggesting that other lipid species such as PE might fulfill the function exerted by cholesterol in mammalian cells. Considering the comparable replication kinetics of ZIKV in mosquito cells cultured in DL-FBS and FBS, we wondered whether DL-FBS cell-grown ZIKV remained infectious for VeroE6 cells. We collected Aag2 culture supernatants at 96 h.p.i. and used it to infect VeroE6 cells. We found that cholesterol-lacking ZIKV was not impaired in its ability to infect VeroE6 cell cultures, arguing that cholesterol from the host cell is decisive for productive infection in mammalian cells. Consistently, when we treated VeroE6 cells for 1 h with MβCD prior to ZIKV infection, we observed a significant reduction in ZIKV-positive cells (Supplementary Fig. 8b, c). This result suggests that the cholesterol content of the host cell target membrane rather than the lipid composition of the virus itself defines ZIKV infection capability.

Given that cholesterol is a priori not required to infect mosquito cells, we next analyzed the ability of CARC2 mutants to infect and replicate in Aag2 cells. Cells were inoculated with Huh7-Lunet grown ZIKV using equal numbers of viral genomes per cells (~60 genome copies) for WT and the CARC2 mutants to avoid confounding effects caused by differences in specific infectivity (Supplementary Fig. 6d). Infected Aag2 cells were harvested in 24-h intervals over a period of 96 h and virus replication kinetics were analyzed by qRT-PCR. We observed well-comparable growth curves between ZIKV WT and CARC2 mutants in Aag2 cells (Supplementary Fig. 8d). Thus, despite differential incorporation of lipids into the virus envelope[32–34], mutations affecting the CARC2 motif in the M-domain of prM did not attenuate ZIKV in the insect host.

### Mutations in the CARC3 motif of ZIKV prM impair virion assembly

While phenotypes of CARC2 mutants were highly dependent on cellular cholesterol levels of target cells, the replication cycle of CARC3 mutants was severely affected independent of cellular cholesterol levels. To define the step of the ZIKV life cycle affected by mutations in the CARC3 motif, we examined the release of virus particles from Huh7-Lunet cells transfected with mutant and WT RNA genomes. Titers of infectious virus particles released into cell culture supernatants were determined by TCID50 assay while the number of virus particles was assessed by measuring viral RNA in the supernatant using RT-PCR and agarose gel electrophoresis (Fig. 6a, b). Relative to WT, both the titer of infectious virus and amounts of viral RNA in the supernatant were drastically reduced for the two ZIKV CARC3 mutants. To discriminate whether this reduction was due to impaired virus assembly or reduced virion egress, amounts of intracellular infectious virus particles were quantified by TCID50 assay (Fig. 6c). We found that intracellular infectivity titers of the CARC3 mutants were decreased throughout the observation period. At 96 h post transfection, titers of the mutants were 1,000-10,000-fold lower than WT, arguing for an assembly defect. To corroborate our assumption, we transfected Huh7-Lunet T7 cells with ZIKV prM/E VLP constructs containing the CARC3 mutations. After 18 h, culture supernatants were collected, proteins contained therein were concentrated and analyzed by western blot (Fig. 6d). Our result shows that also in the ZIKV VLP system the CARC3 mutations reduced E, prM, and M protein amounts released from transfected cells relative to WT. These results supported the notion that CARC3 mutants had an assembly defect. To define the affected step in the viral life cycle more precisely, we used correlative light-electron microscopy (CLEM). To this end, we employed a ZIKV reporter cell line stably expressing an ER-resident GFP sensor as previously described for DENV[35] (Supplementary Fig. 9a). Upon ZIKV infection, the GFP is cleaved off the engineered ER-anchor by the viral NS2B/NS3 protease with GFP trafficking into the nucleus (Supplementary Fig. 9b). In this way, we identified ZIKV-positive cells at 72 h post transfection by light microscopy and using a CLEM set-up (Supplementary Fig. 9c), examined the very same cells by transmission electron microscopy (TEM). In both WT and CARC3 mutant-transfected cells, we detected the viral replication compartments (termed vesicle packets (VPs)) in the lumen of the rough ER (Fig. 6e), convoluted membranes, and zippered ER structures, all being well-reported membrane rearrangements induced in flavivirus infected cells[36]. In addition, in WT virus-transfected cells ~169 virus particles per cell were detected (Fig. 6f). These virions had an average diameter of 44 ± 6 nm (Fig. 6g) and resided within the lumen of the ER (Fig. 6e) and *trans*-Golgi network (Supplementary Fig. 9d). Importantly, virus particles were rarely detected for CARC3 mutant-transfected cells (Fig. 6e; Supplementary Fig. 9d). Although structures resembling virus particles were observed for the CARC3 mutants (mean diameters similar to WT) (Fig. 6g), virion abundance was significantly lower as compared to WT (Fig. 6f). In contrast, the mean number of VPs counted within transfected cells was well comparable between ZIKV CARC3 mutants and WT (Fig. 6f), consistent with their comparable RNA replication capacity. Taken together, these results suggest that CARC3 mutants, which have reduced association with cholesterol, impair the assembly of infectious virus particles.

### The CARC3 motif in prM recruits capsid to virus assembly sites

To exclude the possibility that mutations within the CARC3 motif of the M-domain affect prM release from the polyprotein or prM cleavage by furin, lysates of Huh7-Lunet cells transfected with WT or CARC3 mutant genomes were analyzed by western blot (Supplementary Fig. 10a). Both the abundance of ZIKV proteins and polyprotein processing were comparable between WT and CARC3 mutants (Supplementary Fig. 10b). However, for the CARC3 mutants, we observed an accumulation of the shorter (mature) form of capsid protein, relative to the longer "membrane-anchored" (incompletely processed) form (Fig. 7a, b). This result was consistent with reduced colocalization events between E proteins and capsid (Fig. 7c, d). Notably, measuring

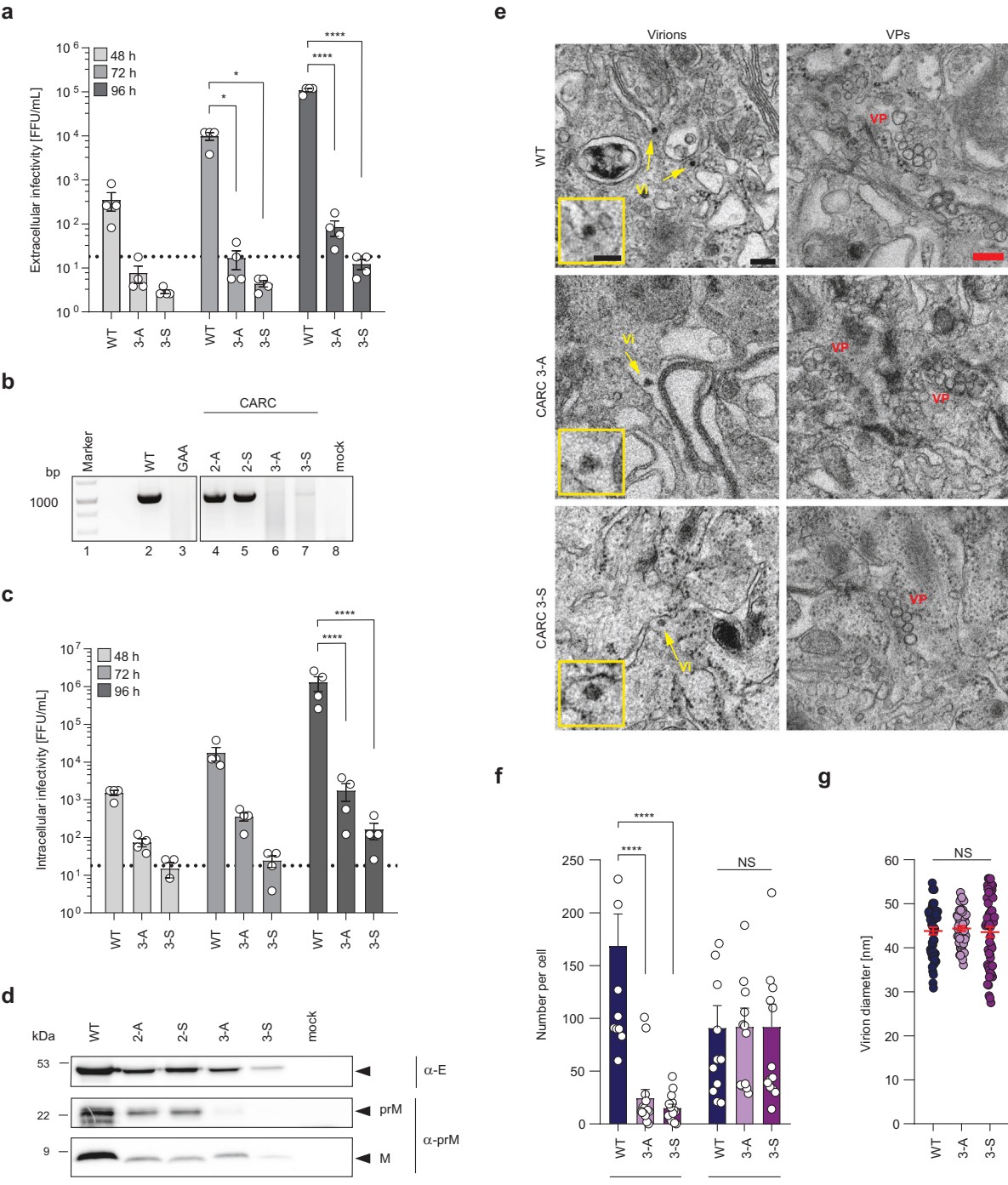

**Fig. 6 | Mutations in the CARC3 domain affect virus assembly. a** Huh7-Lunet cells were transfected with in vitro transcripts of synZIKV-H/PF/2013 and supernatants were collected at 48, 72, and 96 h.p.t. Titers of extracellular infectious virus were determined by TCID50 assay ($n = 4$). Data are mean ± SEM. Two-way ANOVA with Fisher's LSD test, *$P < 0.05$, ****$P < 0.0001$. (WT vs 3-A, $P = 0.0391$; WT vs 3-S, $P = 0.0389$). **b** At 96 h.p.t. total RNA was extracted from supernatants (**a**). Viral RNA contained therein was reverse transcribed and an ~1,000 nucleotides long region within the structural protein coding sequence was amplified by PCR. The non-replicative NS5 mutant (GAA) served as negative control (lane 3). DNA size maker is shown in lane 1 ($n = 2$). **c** Intracellular infectivity was assessed by TCID50 assay. Cells from **a** were scraped off the plate and intracellular virus particles were recovered by repeated freeze-thaw cycles. Clarified supernatants were subjected to TCID50 assay ($n = 4$). Data are mean ± SEM. Two-way ANOVA with Fisher's LSD test, ****$P < 0.0001$. **d** Mutations in the CARC3 motif impair VLP formation. Huh7-Lunet T7 cells were

transfected with ZIKV prM/E VLP constructs. At 18 h.p.t. supernatants were collected and secreted proteins contained therein were concentrated and analyzed by western blot using ZIKV-specific antibodies. Molecular weights are indicated on the left (in kilodalton; kDa) ($n = 2$). **e** Cells expressing a ZIKV-cleavable GFP reporter protein were transfected with in vitro transcripts of synZIKV WT or the CARC3 mutants. After 72 h, transfected cells were processed for CLEM analysis to identify transfected cells. Representative images of up to 15 transfected cells per condition are shown. Scale bar: 200 nm. Zooms of virions are given as inserts. Scale bar in inserts: 100 nm. Vi: virions. Vesicle packets (VP) were recorded as marker for ZIKV replicating cells. **f** Numbers of virions and VPs were quantified for WT ($n = 12$ cells) and CARC3 mutants ($n = 15$ cells). Data are mean ± SEM. $n = 3$ independent experiments. Two-way ANOVA with Dunnett's test, ****$P < 0.0001$. NS not significant. **g** Diameter of identified virions from **e**. Data are mean ± SEM. Kruskal-Wallis with Dunn's test. NS not significant.

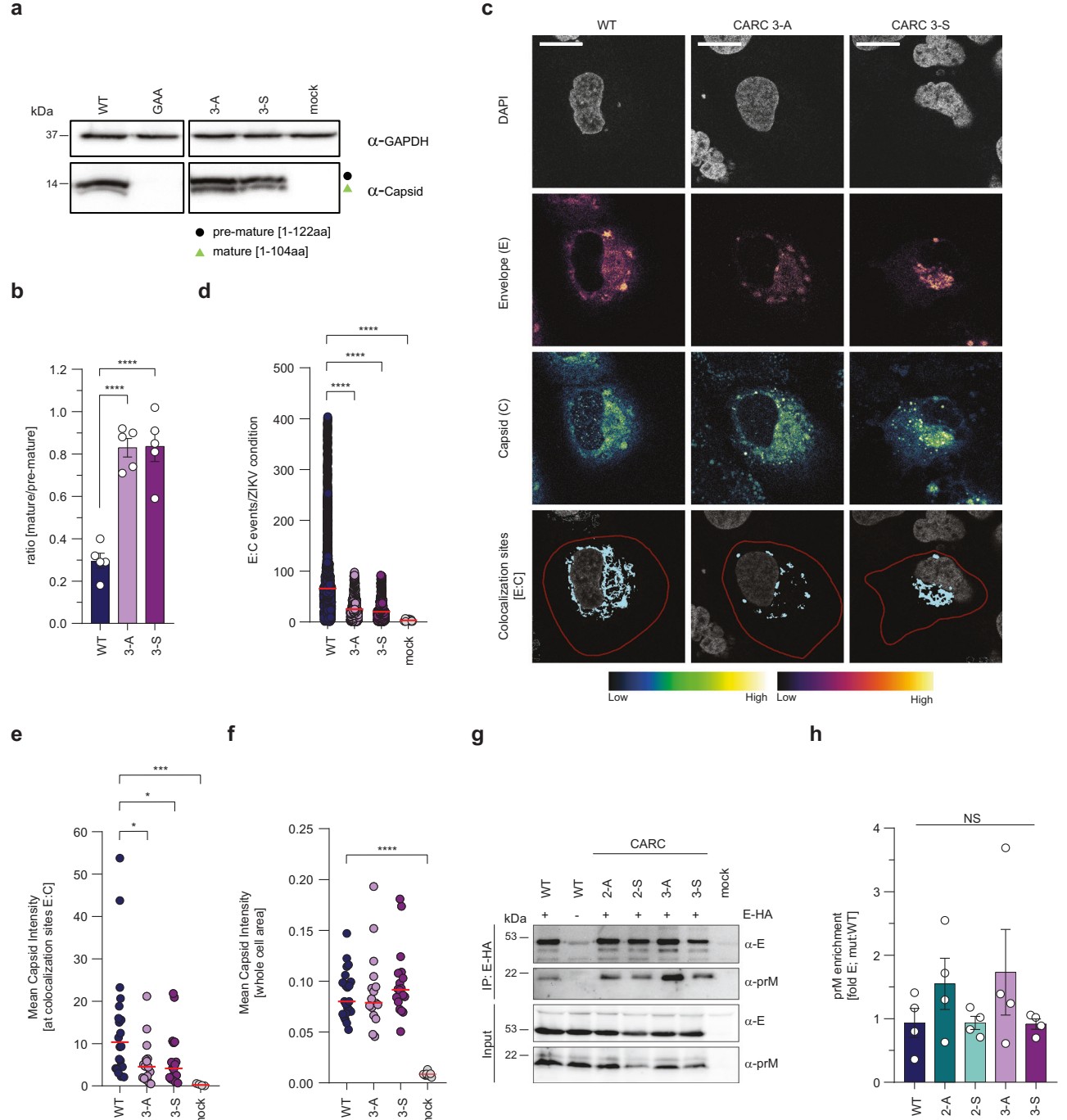

**Fig. 7 | Cholesterol associated with the CARC3 motif in the M-domain and its role in recruiting capsid protein to virus assembly sites. a** Lysates of transfected Huh7-Lunet cells were collected at 48 h.p.t. and subjected to western blot analysis using a capsid-specific antibody. GAPDH served as loading control. The two capsid cleavage products are marked on the right, molecular weights of proteins are indicated on the left (in kilodaltons; kDa). $n = 5$ independent experiments. **b** Capsid protein cleavage products were quantified by densitometry of western blots ($n = 5$). Data are mean ± SEM of the ratio mature/pre-mature proteins. One-way ANOVA with Dunnett's test, ****$P < 0.0001$. **c** Subcellular localization of capsid proteins was assessed by immunofluorescence microscopy. Transfected Huh7-Lunet cells were PFA fixed at 48 h.p.t. and stained for E and capsid. Signal intensities of E and C are indicated on the bottom. Representative micrographs are shown ($n = 2$). Scale bar: 20 μm. **d**–**f** Micrographs from **c** were analyzed using the cell profiler software package. **d** Numbers of C–E colocalization events + median. Two-tailed Mann-Whitney test, ****$P < 0.0001$. **e** Mean capsid signal intensities + median at C–E colocalization sites. Two-tailed Mann-Whitney test, *$P < 0.05$, ***$P < 0.001$. (WT vs 3-A, $P = 0.0112$; WT vs 3-S, $P = 0.0127$; WT vs mock, $P = 0.0001$). **f** Mean capsid signal intensities within whole cells + median. Two-tailed Mann-Whitney test, ****$P < 0.0001$. NS not significant. **g** Mutations affecting the two CARC motifs do not affect E-prM interaction. HEK 293 T cells were transfected with constructs encoding either the WT prM/E-HA proteins or CARC2 and CARC3 mutations. An untagged prM/E construct was used as specificity control. After 18 h cells were lysed and whole cell lysates were subjected to HA-specific immunoprecipitation. Captured HA-E and co-captured prM proteins were analyzed by immunoblotting using E- and prM-specific antibodies. $n = 4$ independent experiments. **h** Relative prM binding to E-HA was calculated by densitometry of the signals in **g**, normalized to input signals. Data are means ± SEM of the prM eluate/input signal ratio, normalized to the respective E signal ($n = 4$). One-way ANOVA with Dunnett's test. NS not significant.

the mean intensities of capsid signals at E−C colocalization sites revealed a significant reduction for both CARC3 mutants (Fig. 7e). These differences were not due to different total C protein signals, as they were well comparable between WT and the two mutants (Fig. 7f) consistent with our results obtained by western blot (Supplementary Fig. 10b). Furthermore, former studies have shown that capsid needs to associate with lipid droplets (LDs) to facilitate virion assembly[5,6]. Therefore, we also quantified total numbers of colocalization events between capsid and LDs (Supplementary Fig. 11a, b) and observed a significant reduction of colocalization events for both CARC3 mutants (Supplementary Fig. 11b). The observation that C-LD colocalization events were reduced in CARC3 mutant transfected cells, prompted us to estimate the total number of LDs per cell profile (Supplementary Fig. 11c). Our result shows that the number of LDs was significantly lower when compared to mock cells, confirming previous reports that during ZIKV infection LDs are consumed[37]. Thus, the observed reduction is not due to defects of capsid association with LDs but rather that LDs are consumed over time as they serve as a source for lipids and energy. In this context, when we measured the mean intensities of capsid signals at LD−C colocalization sites we found no significant difference between CARC3 mutants and WT (Supplementary Fig. 11c). These results suggest that both CARC3 mutant capsids can still associate with LDs and as infection progresses dissociate from LDs to become available as NC building blocks, but formed NCs might be insufficiently incorporated into virions, consistent with the increased level of intracellular mature capsid.

To determine whether the introduced CARC3 mutations affect the subcellular distribution of prM and E proteins, immunofluorescence microscopy was performed. Huh7-Lunet cells were transfected with individual CARC3 mutants and 48 h later, prM, E, or reticulon-3 (RTN3) were detected (Supplementary Fig. 12a, c). For all ZIKV-positive cells, prM was distributed throughout the cytoplasm with occasional cytoplasmic clusters, possibly corresponding to sites of convoluted membranes. Colocalization analysis of prM-E proteins (Supplementary Fig. 12b) and prM-RTN3 proteins (Supplementary Fig. 12d) revealed similar co-occurrence coefficients for WT and CARC3 mutant transfected cells, indicating no effect of CARC3 mutations on subcellular distribution of prM and E.

Since the interaction between prM and E is crucial for virion formation, we next addressed the question whether the mutations within the CARC3 motif disrupt this interaction by using co-immunoprecipitations. To this end, HEK 293 T cells supporting high-level expression, were transfected with constructs encoding prM and HA-tagged E proteins and pull-down of HA-E was performed to monitor coprecipitation of prM. As shown in Fig. 7g, h, efficiency of prM−E coprecipitation was well comparable between WT and the two CARC3 mutants. Taken together, our data confirm that mutations within the CARC3 motif affect neither prM cleavage and maturation, nor the subcellular localization of prM proteins and their colocalization with E. Instead, we observed reduced colocalization of E proteins with capsid arguing that cholesterol association of the M-domain in prM contributes to the recruitment of capsid to assembly sites.

## Discussion

In this study, we employed a chemo-proteomics approach and found that the structural protein prM, via its membrane-resident M-domain, can interact with cholesterol leading to incorporation of this lipid into virus particles (Fig. 1). Using bioinformatics analysis together with site-directed mutagenesis and functional characterization of prM mutants (Fig. 2), as well as atomistic MD simulations (Fig. 3), we identified two cholesterol-binding motifs located within the two transmembrane segments of the M-domain. Although amino acid substitutions within the CARC2 and CARC3 motif greatly reduced cholesterol occupancy (Fig. 2; Supplementary Fig. 3; and Fig. 3), only small conformational changes were observed, which can be described as rotation of TMS2

with respect to TMS3 (Supplementary Movie 1). By using three different cell lines, that differ greatly in their cholesterol content (Fig. 5), we were able to show that (1) cholesterol is a host dependency factor with respect to mammalian but not mosquito host cells (Fig. 5 and Supplementary Fig. 8), (2) a high cholesterol content environment rescues the infection defect of ZIKV CARC2 mutants (Fig. 5 and Supplementary Fig. 6), and (3) disruption of M−cholesterol interaction impairs ZIKV particle production (Fig. 6). Hence, we conclude that the M-domain is a cholesterol interactor, having two cholesterol binding motifs within its two transmembrane helices that exert two different functions during the ZIKV replication cycle.

Since formation of endocytic vesicles requires cholesterol[38], treatments with cholesterol solubilizing drugs such as MβCD exhibits antiviral activity by hampering virus infection[23,24]. Conversely, viral fusion activity is enhanced when cholesterol is present within target membranes as shown for Semliki Forest virus (SFV)[39] and WNV[10]. Mechanistically, cholesterol promotes fusion of the viral envelope with the endosomal membrane by creating space between bulky head groups within the phospholipid bilayer, thus driving envelope fusion loop insertion as shown for the Rift Valley Fever Virus (RVFV)[40]. In addition, cholesterol contributes to lipid mixing of the envelope and endosome membranes as reported for influenza A virus (IAV)[41]. As shown here for the CARC2 mutants, also ZIKV entry into host cells very much depends on cholesterol. Infectivity of these mutants for the "low-cholesterol" VeroE6 cells was profoundly reduced, but not affected in "high-cholesterol" cells (Fig. 5). We could show that pharmacologically lowering cholesterol levels in host target membranes using MβCD, leads to premature virus fusion in proximity to the PM (Supplementary Fig. 7), confirming the previous report[39] highlighting that apart from the acidic pH also the lipid composition, in particular cholesterol, is a determinant for the initiation of viral envelope fusion in mammalian cells. Studies of SFV fusion have highlighted that removal of cholesterol from liposomes abrogated the delivery of viral RNA into the lumen of liposomes[39]. While fusion per se was not affected for the CARC 2-S mutant (Fig. 5), our data together with previous findings suggest that premature fusion hampers the delivery of the viral genome into the cytoplasm, and thus productive infection. Since we employed CARC2 mutants produced in high-cholesterol cells, i.e., virus particles loaded with cholesterol, we assume that the primary defect caused by the CARC2 mutations is insufficient interaction with cholesterol in the endosomal membrane of target cells rather than cholesterol "loading" of the viral envelope via prM. Considering previous studies[39,41], our results therefore suggest that M might mediate lipid mixing, a process required to form the fusion pore. However, further studies are needed to decipher whether ZIKV CARC2 mutants are halted in a hemifusion conformation preventing the delivery of the viral genome into the cytoplasm.

ZIKV infection and replication cycle do not depend on cholesterol in mosquito cells, because the CARC2 mutants were as fit as WT in these cells (Supplementary Fig. 8). As insects are deficient in de novo biosynthesis of cholesterol and rely on dietary sources to complete their gonotrophic cycle[42], our data argue that entry processes are coordinated by different lipid species depending on the host. In fact, when comparing the organization of phospholipids within the PMs of mammals and insects, the most striking difference is that mammalian PMs are asymmetrical[43] whereas insects' PMs are symmetrical[44]. More specifically, the ratio of PE to phosphatidylcholine is 4-fold higher in insect cells than in mammalian cells[45]. Interestingly, cholesterol and PE were found to serve as modulators of PM fluidity in mammals and insects, respectively[45]. In this context, the interferon-induced trans-membrane 3 protein prevents virus fusion via its interaction with cholesterol, leading to an increase of cholesterol levels at the interphase and stabilizing the hemifusion[46]. Moreover, interference with cholesterol recycling via the Niemann-Pick type C protein prevents ZIKV envelope fusion with the endosome[47].

Infection capabilities of ZIKV CARC2 mutants depended on host cell cholesterol content (Fig. 5). Since cholesterol has antiviral properties[48], our observation that CARC2 mutants fused much faster in low-cholesterol VeroE6 cells than WT might be due to insufficient delay of fusion because of insufficient cholesterol levels. In contrast, in high cholesterol Huh7-Lunet cells this inhibition might suffice to delay envelope-endosome fusion of CARC2 mutants to allow proper uncoating of the virus particles. In any case, our data argue for a narrow range of membrane cholesterol concentration supporting virus entry.

Within the last couple of years, research has provided valuable insights into the molecular mechanism of how ZIKV particles are being formed[49]. It was found that during assembly, the capsid protein interacts with the inner leaflet of the viral envelope via its highly hydrophobic helix α1 residing in the N-terminal region of capsid[5,50]. This interaction occurs preferentially in regions where the transmembrane domains of E and M proteins are located[51]. However, thus far no evidence of direct interaction between capsid and these two proteins has been reported. Alternatively, the interaction might be mediated via lipids such as cholesterol. We note that mutations affecting the CARC3 motif in M dampened assembly and reduced the number of colocalization events between E and capsid while prM–E colocalization was not affected (Supplementary Fig. 12). These results argue for direct prM–E interaction, consistent with our pull-down data (Fig. 7g, h), whereas capsid might associate with prM/E complexes via cholesterol enriched at assembly sites. This enrichment might be mediated by prM associating with cholesterol. Alternatively, cholesterol-rich sites, such as lipid rafts, might pre-exist and via prM association with cholesterol, the prM/E complex is recruited to these sites where it can associate with capsid during assembly. In either case, cholesterol-rich sites would serve as anchoring platform for non-structural and structural proteins involved in virus particle formation. In addition, to this function, cholesterol might also contribute to membrane curvature as the virus buds into the lumen of the ER, similar to what has been proposed for other viruses such as HIV[52] and IAV[53]. The assembly reaction might be facilitated by additional interactions between cholesterol and capsid as well as NS2A and NS2B that also contain potential cholesterol-binding sites in their transmembrane domains.

Cryo-electron tomography has provided atomistic resolution of immature and mature virus particles[51] revealing the incorporation of certain lipids into the envelope[33,34]. However, cholesterol was neither identified, nor reconstructed within the lipid envelope of ZIKV and related flaviviruses. This lack of structural information could be explained by the fact that virions used for cryo-EM structural analyses were produced in mosquito cells and in a low-lipid environment. Here, we demonstrate that ZIKV does not require cholesterol for replication in mosquito cells (Supplementary Fig. 8), but cholesterol is incorporated into the lipid envelope most likely via its interaction with the membrane resident M-domain of prM when ZIKV is grown in mammalian cells (Fig. 1). Additionally, the lack of evidence for cholesterol incorporation into the viral envelope in the cryo-EM studies could be explained by our MD simulations showing that cholesterol binding is dynamic and thus, difficult to capture in the cryo-EM density, unless crosslinked to proteins in proximity, which has not been done in these studies.

In summary, we report two novel functions of the ZIKV M protein playing a role in virus entry and assembly of infectious virus particles. Our data provide novel mechanistic insights into viral protein–lipid interactions and their role in the viral life cycle.

## Methods

### Cell lines and viruses
Huh7-Lunet cells[14], Huh7-Lunet T7 cells[16], VeroE6 (ATCC CRL-1586), HEK 293 T (ATCC CRL-3216), and A549 (ATCC CCL-185) cells were cultured in Dulbecco's modified Eagle medium (DMEM) containing 10% fetal bovine serum (FBS; Sigma-Aldrich), 100 U/mL penicillin, 100 μg/mL streptomycin, and 1% MEM non-essential amino acids (all from Gibco, Life Technologies). Cells were kept at 37 °C with 5% $CO_2$. To maintain a stable expression of the T7 RNA polymerase, zeocin (Invitrogen) was added to the culture medium of Huh7-Lunet T7 cells at a final concentration of 5 μg/mL. The human placental cell line JEG3, kindly provided by Udo Markert, University of Jena, Germany, was maintained in Ham's F-12 Nutrient Mix medium (Gibco, Life Technologies), supplemented with 10% FBS, 100 U/mL penicillin, and 100 μg/mL streptomycin. The mosquito cell line Aag2, kindly provided by Alain Kohl, MRC, Glasgow, was grown in Leibovitz L-15 medium (Gibco, Life Technologies) supplemented with 10% tryptose phosphate broth (TPB, Gibco, Life Technologies), 10% FBS, 100 U/mL penicillin, and 100 μg/mL streptomycin. The C6/36 mosquito cell line (ECACC 89051705) was grown in Leibovitz L-15 medium supplemented with 10% FBS, 100 U/mL penicillin and 100 μg/mL streptomycin. Both mosquito cell lines were kept at 28 °C without $CO_2$.

ZIKV H/PF/2013 (obtained from the European Virus Archive) stocks were prepared by virus amplification in the mosquito cell line C6/36. Extracellular titers were determined by plaque forming units (PFU) assay in VeroE6 cells using an overlay medium containing 1.5% carboxymethylcellulose. synZIKV-H/PF/2013 stocks were prepared by electroporation of Huh7-Lunet cells with in vitro transcripts. Titers of extracellular virus were determined by tissue culture infection dose 50 (TCID50) assays in Huh7-Lunet cells.

### DNA constructs
The reverse genetics system for ZIKV (strain H/PF/2013, GenBank accession number KJ776791.2; synZIKV and synZIKV-R2A) has been previously described[22]. Site-directed mutagenesis of the cholesterol binding motifs within the M coding region of prM of the synZIKV-H/PF/2013 molecular clone was performed by overlap PCR. Briefly, in a two-step PCR reaction, fragments containing the desired mutations were generated. Following gel extraction and purification, fragments were digested with NheI and SphI (New England Biolabs) and inserted into the full-length cDNA of synZIKV- and synZIKV-R2A-H/PF/2013. The prM coding region amplified from WT or mutant ZIKV cDNA was inserted into the pTM backbone. This prM sequence contained the capsid anchor sequence at the 5'end and an HA-tag at the 3'end.

Constructs for the ZIKV VLP system were generated similar to a previous report[15]. Briefly, the prM-E construct was generated by PCR amplification of the prM-E coding region comprising codons 105 to 795, using the Phusion high-fidelity PCR kit (ThermoFisher). After gel extraction and purification, the amplicons were inserted into the pTM vector by use of the NcoI and BamHI restriction sites to allow continued expression of the prM-E polyprotein in Huh7-Lunet T7 cells. All DNA constructs were fully sequenced to ensure their integrity. Primers used for DNA cloning are listed in Supplementary Table 1.

### Generation of the ZIKV reporter cell line
Lentiviral transduction was used to generate a stable cell line expressing the ZIKV reporter construct. Lentiviruses were produced by transfecting HEK 293 T cells with packaging plasmids pCMV-Gag-Pol and pMD2-VSV-G (kind gifts from Didier Trono, EPFL, Lausanne) and the pWPI vector encoding the protease reporter construct. Two days post transfection, lentivirus-containing supernatants were collected and filtered. Huh7-Lunet cells were transduced in the presence of 4 μg/mL polybrene. Cells were selected and maintained in puromycin (1 μg/mL) containing medium.

### Multiple-sequence alignment of prM from different flaviviruses
M-protein sequences of different flaviviruses were aligned using the PRALINE multiple-sequence alignment toolbox[54]. Membrane topology was predicted according to previous reports[55,56].

### In vitro transcription and RNA transfection

Infectious viral RNAs were generated from plasmids pFK_synZIKV-H/PF/2013 and pFK_synZIKV prM$_{mut}$ by in vitro transcription as previously described[22]. For transfection, 10 μg of in vitro transcripts was added to 4E + 06 Huh7-Lunet cells or 6E + 06 VeroE6 cells in 0.4 mL of cytomix buffer (120 mM KCl, 0.15 mM CaCl$_2$, 10 mM potassium phosphate buffer, 25 mM HEPES (pH 7.6), 2 mM EGTA, 5 mM MgCl$_2$, freshly supplemented with 2 mM ATP and 5 mM glutathione), transferred into 4-mm gap cuvettes followed by a pulse at 975 μF and 0.27 kV using the GenePulser apparatus (BioRad). Transfected cells were recovered in pre-warmed DMEM complete, seeded into multi-well cell culture plates, and kept at 37 °C and 5% CO$_2$. At time points specified in the results section, replication kinetics of the recombinant viruses were analyzed.

### Immunofluorescence microscopy

Transfected Huh7-Lunet cells grown on glass coverslips were washed thrice with PBS, fixed with 4% paraformaldehyde (PFA) at 22 °C for 10 min, and permeabilized using 0.2% Triton-X in PBS for 10 min at 22 °C. Samples were blocked in blocking buffer (1% skim milk in PBS) for 1 h at 22 °C, followed by incubation with primary antibodies diluted in blocking buffer for 16 h at 4 °C (primary and secondary antibodies used are given in Supplementary Table 2). After washing thrice with washing buffer (0.01% Tween20 in PBS), samples were incubated with Alexa Fluor 488-, or 568-conjugated secondary antibodies for 1 h at 22 °C. For visualization of lipid droplets, the molecular probe Bodipy-488 (Invitrogen) was used. Samples were mounted on microscope slides using DAPI Fluoromount-G (Invitrogen) mounting media. Images were acquired with either a Nikon Eclipse Ti microscope or a Leica SP8 DLS Laser Scanning Confocal SPIM microscope and analyzed using either the FIJI or the Cell Profiler software package. Adobe Illustrator was used to assemble images into figures.

Determination of capsid intensity at envelope–capsid (E–C), or lipid droplet–capsid (LD–C) colocalization sites was done as described before[57]. In brief, segmentation of capsid, envelope, and lipid droplets was performed using iLAstik pixel classification. This was followed by the masking of identified objects in E–C and LD–C pairs, and measurement of mean intensity of masked objects in a custom-made Cell Profiler script. Significance was calculated and graph was plotted in GraphPad Prism version 8.

### TCID50 assay

One day prior to infection, Huh7-Lunet cells were seeded into 96-well plates at a density of 8E + 03 cells per well. On the day of infection, each supernatant was initially 2-fold diluted, followed by seven 10-fold serial dilutions in a total volume of 200 μL per well. Each supernatant was analyzed in sextuplicate. After 48 h, cells were fixed in methanol at −20 °C for at least 1 h. Plates were briefly airdried, washed once with PBS, and cells were permeabilized using 0.5% Triton X-100 in PBS for 5 min. Plates were washed thrice with PBS, and incubated with the panFlavi anti-Envelope primary antibody for 3 h at 22 °C. Plates were washed thrice with PBS, followed by incubation with HRP-coupled anti-mouse secondary antibodies for 1 h at 22 °C. Positive cells were visualized with a homemade substrate (5 mL Acetatos (0.5 M sodium acetate, 0.5 M acetic acid in H$_2$O), 1.5 mL Carbazol (1.05 mM in methanol), and 20 μL H$_2$O$_2$, per plate, filtered through a 0.45 μm-pore filter). Reaction was allowed for at least 30 min, depending on staining intensity. Substrate was removed and plates were washed with H$_2$O. Using a light microscope (Leica), positive cells were identified, and titers were calculated with a TCID50 calculator (https://www.klinikum.uni-heidelberg.de/Downloads.126386.0.html).

### Photoaffinity crosslinking with cholesterol

ZIKV proteins were probed for cholesterol interaction in living mammalian cells as previously described[12,13,41,58]. In brief, Huh7-Lunet cells were grown in six-well plates and infected with ZIKV H/PF/2013 (MOI = 10). After 24 h, cells were incubated with 10 μM PAC-cholesterol (Sigma Aldrich: 804657-5MG) in DMEM supplemented with 10% delipidated- (DL)-FBS for 1 h at 37 °C. Cells were washed with cold PBS, treated with ultraviolet light for 5 min at 4 °C, and lysed at 4 °C for 1 h in 100 μL lysis buffer (PBS, 1% Triton X-100, 0.1% SDS, and protease inhibitor cocktail (Sigma Aldrich)). Protein concentrations in clarified lysates were determined via Bradford assay. Equal protein amounts were subjected to Cu$^I$-catalyzed click-chemistry (5 μL of each: 1% SDS in PBS, 25 mM freshly prepared CuSO$_4$ (Sigma-Aldrich) in H$_2$O (final concentration 1.25 mM), 25 mM freshly prepared ascorbic acid (AppliChem) in H$_2$O (final concentration 1.25 mM), 2.5 mM of freshly prepared tris[(1-benzyl-1H-1,2,3-triazol-4-yl)methyl]amine (TBTA, Roth) in DMSO (final concentration 125 μM), and 2.5 mM freshly prepared azide-biotin (Jena Bioscience) in DMSO (final concentration 125 μM) in a total volume of 125 μL. Click reaction was done at 37 °C for 3 h, and proteins were precipitated with methanol at −80 °C for 16 h. After washing, air drying, and reconstituting protein pellets in 0.2% SDS in PBS, affinity purification was performed using Neutravidin beads (ThermoFisher). After 3 h, complexes were washed 10 times with 0.2% SDS in PBS. Proteins were eluted using 2X Laemmli sample buffer (6X buffer: 600 mM Tris-HCl (pH 6.9), 15 mM EDTA, 0.1% (w/v) bromophenol blue, 10% (w/v) SDS, 7.5% (v/v) β-mercaptoethanol, 30% (v/v) glycerol) and captured complexes were analyzed by western blot.

### Quantitative real-time PCR

Total cellular RNA was isolated from cells using the NucleoSpin RNA extraction kit (Macherey-Nagel) according to the manufacturer's protocol. Amounts of ZIKV RNA were determined by either using the PerfeCTa qPCR ToughMix (Quantabio) or the SYBR Green qPCR Supermix (BioRad). For the former, 15 μL of reaction mixes contained 7.5 μL of 2X PerfeCTa mix, 400 nM of each primer, 200 nM of each probe, and 3 μL of extracted total RNA. For absolute quantification, a 10-fold serial dilution series ($10^3$ to $10^9$ copies/μL) of ZIKV transcripts were added as standard to each plate. Each sample was analyzed in triplicates. GAPDH or RPS17 were used for normalization in mammalian and mosquito cells, respectively. For the latter, cDNA was synthesized using the High-Capacity cDNA Reverse Transcription kit (Applied Biosystems) and two-step qRT-PCR was performed using the iTaq Universal SYBR Green mastermix (BioRad). For both protocols, quantitative real-time PCR (qPCR) were performed on a CFX96 Real-Time System (BioRad). Data were analyzed by using the ΔΔCT method as previously described[59]. Sequences for primers and probes are supplied in Supplementary Table 3.

### Virus replication assays by *Renilla* luciferase assay

*Renilla* luciferase activity was determined as previously published[60]. Briefly, transfected cells were lysed at indicated time points by addition of 100 μL luciferase lysis buffer (25 mM Glycine-Glycine (pH 7.8), 15 mM MgSO4, 4 mM EGTA, 10% (v/v) glycerol, 0.1% (v/v) Triton X-100, freshly added 1 mM DTT). Lysates were stored at −20 °C until further processed. Luciferase activity was assessed with a tube luminometer (Berthold, Technologies). For each sample, 20 μL of lysate was mixed with 50 μL of freshly prepared luciferase assay buffer (25 mM Glycine-Glycine (pH 7.8), 15 mM K$_4$PO$_4$ buffer (pH 7.8), 15 mM MgSO$_4$, 4 mM EGTA, and 1.42 μM coelenterazine).

### Viral RNA protection assay

Supernatants were harvested at 96 h.p.t. from mock- and ZIKV-transfected Huh7-Lunet cells and filtered through a 0.45 μm pore filter. After filtration, 250 μL of supernatant was mixed with 5 μL of RNase A/T1 (ThermoFisher) and incubated at 37 °C for at least 1 h to remove contaminating transfected input RNA. Viral RNA was extracted by TRIzol:chloroform extraction and isopropanol precipitation. Purified viral RNA was reverse-transcribed using an E-gene-specific reverse

primer and the Superscript III reverse-transcriptase polymerase (ThermoFisher). Secretion of viral RNA was assessed by amplifying the structural genes using the High-fidelity Phusion polymerase (ThermoFisher) with gene-specific primers. Amplicons were resolved by 1% agarose gel electrophoreses and recorded using the GelDoc software (INTAS). Primers used for detection of extracellular viral RNA are supplied in Supplementary Table 3.

## ZIKV VLP secretion assay

Supernatants were collected 18 h.p.t. from mock- and ZIKV VLP-transfected Huh7-Lunet T7 cells and proteins contained therein were precipitated using 18% volume of polyethylenglycol (PEG). Precipitation was allowed for 18 h at 4 °C. On the next day, proteins were pelleted via centrifugation at 10,000×$g$ for 1 h (4 °C), followed by two washing steps using PBS and a 100 K Amicon filtration device (3500×$g$ for 15 min at 4 °C). Retained proteins were transferred to a fresh 1.5 mL reaction tube and proteins were precipitated at −80 °C using methanol for ~18 h. On the next day proteins were pelleted by centrifugation (16,000×$g$ for 10 min at 4 °C), and pelleted proteins were reconstituted in equal volumes of 2X Laemmli sample buffer at 95 °C for 15 min. Proteins were resolved by SDS-PAGE and analyzed by western blotting.

## PAC-cholesterol incorporation assay

Huh7-Lunet T7 cells were transfected with the ZIKV VLP expression construct and 18 h later, cells were fed with 10 μM of the PAC-cholesterol probe for 4 h. Thereafter, supernatants were collected, and proteins contained therein were concentrated using 18% volume PEG. After protein precipitation at 4 °C for 18 h, proteins were pelleted by centrifugation at 10,000×$g$ for 1 h (4 °C). Pelleted proteins were washed twice with PBS using a 100 K Amicon filtration device (allowing the removal of PEG and excess of PAC-cholesterol). Retained proteins were subjected to UV irradiation to cross-link PAC-cholesterol to proteins in proximity followed by click chemistry using an azide fluorescent dye (JenaBioscience: CLK-1182). Upon click chemistry, proteins were precipitated in methanol at −80 °C for up to 18 h. Proteins were pelleted on the next day by centrifugation (16,000xg for 10 min at 4 °C) followed by reconstituting proteins in a total volume of 60 μL 1X Laemmli sample buffer. Total proteins contained in equal volumes were resolved by SDS-PAGE and gels were imaged using a fluorescent imager (LICOR) to record fluorescently labeled cholesterol-protein complexes. Proteins were transferred onto PVDF membranes and analyzed by western blotting.

## SDS-PAGE and western blotting

Transfected cells were lysed at 48 h.p.t. using RIPA lysis buffer (50 mM Tris-Hcl (pH 6.9), 150 mM NaCl, 1% (w/v) sodium deoxycholate, 1% (v/v) Nonident P-40, 0.1% (w/v) SDS) supplemented with 1 μL benzonase (Merck Millipore) followed by denaturation at 95 °C for 5 min. Equal amounts of total proteins (whole cell lysates) or equal volumes of secreted proteins were resolved by SDS-PAGE and transferred to polyvinylidene difluoride (Cytiva) membranes that were blocked with PBS containing 5% skim milk. After >1 h, membranes were incubated in PBS containing 1% skim milk and primary antibodies overnight at 4 °C. On the following day, membranes were washed and incubated with HRP-conjugated anti-mouse or anti-rabbit secondary antibodies for 1 h at 22 °C. Membranes were imaged using the ECL system (Perkin Elmer) and the ChemoStar 6.0 imaging software (INTAS). Images were analyzed using the LabImage 1D (INTAS) and cropped using the FIJI software package. Primary and secondary antibodies used are listed in Supplementary Table 2.

## Coimmunoprecipitation of E and prM proteins

To assess whether mutations within the M-domain affect the interaction with envelope, co-immunoprecipitation assays were performed.

Briefly, HEK 293 T cells were seeded into 10-cm diameter dishes at a density of 5E + 06 cells per dish. One day post seeding, cells were transfected with pcDNA3.1 vector encoding the gene of interest. For one 10-cm dish, 15 μg of plasmid DNA was mixed with 800 μL of serum-reduced OptiMEM (Gibco, Life Technologies) and 45 μL of PEI (Polyplus Transfection) by vortexing. Complex formation was allowed for 18 min at ~22 °C and transfection mix was added to the cells in a dropwise manner. At 18 h.p.t. cells were washed once with PBS, scraped off the plate, and transferred into 2 mL reaction tubes followed by centrifugation at 3000×$g$ for 5 min. Supernatants were aspirated and cell pellets were lysed in 200 μL of DDM lysis buffer (0.5% (w/v) DDM, 150 mM NaCl, 20 mM Tris-HCl (pH 6.9), 1 tablet cOmplete protease inhibitor (Roche)) for 1 h on ice. Clarification of lysates was achieved by centrifugation for 8 min at 16,000×$g$ and 4 °C. Clarified supernatants were transferred to fresh 1.5 mL reaction tubes and protein concentrations were determined via Bradford assay. Cell lysates were normalized to the sample with lowest total protein concentration. 20% of total normalized cell lysate was saved as input and the remaining lysate was mixed with HA-specific agarose beads slurry (Sigma-Aldrich). After 16 h at 4 °C and gentle rotating, resin beads were washed thrice with DDM lysis buffer followed by two wash steps with PBS. Samples were eluted using 2X Laemmli sample buffer and boiling at 95 °C for 15 min. Efficacy of prM−E co-immunoprecipitation was determined by western blot.

## Binding assays

Virus binding assays were performed as described before[61]. Briefly, VeroE6 cells were seeded into 24-well plates one day prior to infection. On the day of infection, cells were precooled on ice for 10 min, followed by 90 min incubation on ice with equal genome copy numbers in binding buffer (DMEM (pH 7.4) containing 0.2% BSA, 2 mM MgCl₂, 1 mM CaCl₂). Cells were washed three times with ice-cold PBS and lysed in RNA lysis buffer (Macherey-Nagel). Virus binding was quantified by qPCR.

## Neutralization assays

To test whether mutations in the M-domain cause structural changes in the ectodomain II of the E protein, neutralization assays were performed. Briefly, VeroE6 cells were seeded into 24-well plates one day prior to infection. On the day of infection, ZIKV WT and CARC2 mutants were incubated with the E-neutralizing antibody 4G2 for 2 h at 37 °C. Thereafter, cells were precooled on ice for 10 min, followed by 90 min incubation on ice with equal genome copy numbers of pretreated ZIKV. As control, binding assays with ZIKV not treated with the antibody were performed in parallel. Cells were washed three times with ice-cold PBS and lysed in RNA lysis buffer (Macherey-Nagel). Virus binding and neutralization were quantified by qRT-PCR.

## R18-labeling of ZIKV and fusion assays

The labeling of ZIKV with octadecyl rhodamine (R18) was done as previously described[25]. In brief, up to 10E + 06 virions of WT and CARC 2-S mutants were incubated with 1 μM rhodamine B-octadecyl ester (Sigma) for 2 h at RT in light-protected conditions. Following this, the unincorporated dye was removed by ultrafiltration using PD-MiniTrap G-25 filters as per manufacturer's guidelines (Cytiva #28-9180-07). Infectivity of labeled virions was determined relative to mock labeled viruses using double-stranded RNA antibody staining at 24 h post infection (MOI = 2).

The fusion assays were performed on VeroE6 cells seeded into 96-well imaging plates (Cellvis) for 16 h followed by infection with ZIKV-R18 WT and CARC 2-S mutants (MOI = 2) at 37 °C for 2 h. Cells were fixed with 4% PFA, stained with DAPI and Alexa fluor 488 conjugated wheatgerm agglutinin (WGA; Thermo W11261), and imaged using a Zeiss CellDiscovery 7 microscope.

## ZIKV fusion upon drug treatment

VeroE6 cells were seeded into 96-well plates one day prior to ZIKV infection and drug treatment. To investigate the impact of pharmacologically lowered cholesterol levels on ZIKV fusion capability, VeroE6 cells were treated with various concentrations of methyl-β-cyclodextrin (MβCD) for 1 h followed by infection with R18-labeled ZIKV WT (MOI = 2). For maintaining a low cholesterol environment, treatment and infection were performed in DMEM supplemented with 10% delipidated FBS. Virus fusion was allowed at 37 °C for 2 h. Thereafter, cells were PFA fixed, and R18 signals were recorded using fluorescence microscopy. Cell viability was determined using the CellTiter Glo cell viability kit (Promega).

All the images for virus fusion experiments were acquired as a Z-stack of 0.5 μm thickness and maximal-projections were used for further analysis.

Quantification of viral fusion events was performed using the Cell Profiler software package (Broad Institute). In brief, puncta of fused virions and cell boundaries were segmented using R18 and WGA fluorescence profiles, respectively, and virus masks were then associated to each cell using the Cell Profiler "RelateObjects" module. The graphs were plotted in GraphPad Prism version 8.

## ZIKV infection upon drug treatment

Huh7-Lunet cells were seeded into 96-well plates one day prior to ZIKV infection and drug treatment. To test the ability of ZIKV to infect cholesterol-depleted Huh7-Lunet cells, they were treated with various concentrations of methyl-β-cyclodextrin (MβCD) for 3 h followed by infection with ZIKV (MOI = 2). For maintaining a low cholesterol level, treatment and infection were performed in DMEM supplemented with 10% delipidated FBS. Virus infection was allowed at 37 °C for 1 h. Thereafter, inoculum was replaced by fresh DMEM supplemented with 1.5% FBS. After 24 h, cells were PFA fixed, and infection was assessed by immunofluorescence microscopy. Cell viability was determined using the commercially available CellTiter Glo cell viability kit (Promega).

Quantification of percent Zika envelope-positive cells was performed as described before[62]. In brief, nuclei were segmented as primary objects and cellular boundary was identified by expanding the nuclear area by 6 pixels in a custom-made Cell Profiler script. Envelope intensity was measured, and cells were classified as infected or non-infected based on a Random-Forest classifier in Cell Profiler Analyst using top 5 features. Significance was calculated and graph was plotted in GraphPad Prism version 8.

## Cholesterol quantification in cells

The cholesterol content of mammalian and mosquito cells was determined using the Amplex Red Cholesterol assay kit (Invitrogen). For this, 1E + 06 cells were washed with PBS and pelleted by centrifugation at 3000x$g$ for 5 min. Cell pellets were resuspended in 100 μL lysis buffer (PBS, 1% Triton X-100, 0.1% SDS) and resuspended by pipetting up and down. Cholesterol content was determined according to the manufacturer's protocol, using the plate reader CLARIOstar$^{Plus}$ (BMG Labtech) at 540-nm excitation and 590-nm emission wavelengths.

## Correlative light-electron microscopy

Transfected Huh7-Lunet ZIKV reporter cells were seeded into MatTek gridded dishes at a density of 4E + 04 cells per dish. At 72 h.p.t. positions of positive cells were recorded using transmitted light with a differential interference contrast configuration and in fluorescence using a GFP filter cube to detect the reporter signal, followed by fixation with 2.5% glutaraldehyde/1% PFA/2% sucrose in 50 mM sodium cacodylate buffer (CaCo) (50 mM CaCo, 50 mM KCl, 2.6 mM MgCl$_2$, 2.6 mM CaCl$_2$, pH 7.4) for 30 min at -22 °C. Following three washes with 50 mM CaCo buffer, cells were treated with 2% osmium tetroxide (Electron Microscopy Science) in 50 mM CaCo for 40 min at 4 °C. After three washes with EM

grade H$_2$O, samples were treated with 0.5% uranyl acetate for >16 h at 4 °C. After 30 min of washing with H$_2$O, samples were progressively dehydrated using increasing concentrations of ethanol (50% to 100%). Following dehydration, samples were infiltrated using a polymerizing Epon resin (Roth) for 48 h at 60 °C. Embedded cells were trimmed, sectioned into 70-nm-thick slices using an Ultracut UCT microtome (Leica) and a diamond knife (Diatome), and collected on slot grids (Plano). Counterstaining was accomplished using 3% uranyl acetate in H$_2$O for 5 min and 2% lead citrate (Reynolds) in water for 2 min. Cells were visualized with a JEOL JEM-1400 transmission electron microscope (Jeol Ltd., Tokyo, Japan). Virion numbers were determined manually by analyzing three consecutive sections, corresponding to a total thickness of 210 nm throughout the cytoplasm of 5 cell profiles per condition ($n$ = 3) using the FIJI software package.

## Atomistic molecular dynamics simulations

The cryo-EM structure of the mature ZIKV (PDB ID: 5IRE) shows two E-M heterodimers. The inner core of this heterodimer consists of a dimer of M protein. Therefore, we used M-protein dimers as basis of our atomistic molecular dynamics simulations. A WT M-protein dimer was first isolated from the complete biological assembly using the cryo-EM structure (PDB ID: 5IRE)[20]. The following mutation pairs were modeled using Modeller[63] based on the wild-type dimer: CARC 2-A, CARC 2-S, CARC 3-A, CARC 3-S. Each protein form was then embedded separately into membranes composed of a binary mixture of 1-palmitoyl-2-oleoyl-glycero-3-phosphocholine (POPC) and cholesterol (CHOL). The following lipid mole percent ratios (mol%:mol% CHOL:POPC) were considered: 0:100, 10:90, 20:80, 30:70. The positioning of the protein in the membrane was determined using the Positioning of proteins in membranes (PPM) webserver[64]. We note that the M-protein has a shorter hydrophobic length that can span most biological membranes (Fig. 3a, b). Altogether, 20 different systems (5 protein variants × 4 membrane compositions) were constructed. For each system, 10 simulation repeats were performed, each 1 μsec long. All simulation systems were constructed using CHARMM-GUI[65]. For each system, the protein was embedded in a hexagonal membrane. The numbers of lipids and water were adjusted to obtain a sufficiently large hexagonal prism box with a base edge of ~85 Å and a height of ~105 Å. Each membrane-protein system was supplemented with counter-ions to neutralize it after which 150 mM KCl was added to the system. As force fields, we used the Charmm36(m) force field for lipids[66] and the protein[67], the TIP3P[68] model for water molecules, and a compatible parameter set for the ions[69]. WYF[70] parameters were used for improved cation-π interactions and the Hydrogen Mass Repartitioning (HMR) method[71] was applied to achieve a larger integration time step.

All simulations were carried out using GROMACS 2021[72]. Before the production runs, each simulation system was first subjected to the equilibration protocol provided by CHARMM-GUI with subsequent simulations in the NVT and NpT ensembles. For the production runs, the equations of motion were integrated using a leap-frog algorithm with a 4-fs time step, as made possible by HMR. The LINCS[73] algorithm was used to constrain all bond lengths. Periodic boundary conditions were used along all three dimensions. For electrostatic interactions, a real space cut-off of 1.2 nm was used. Long-range electrostatic interactions were computed using the fast smooth Particle-Mesh Ewald[74] (SPME) method with a Fourier spacing of 0.12 nm and a fourth-order interpolation. For the van der Waals interactions, a Lennard-Jones potential with a force-switch between 1.0 and 1.2 nm was used. All production simulations were performed in the NpT ensemble. The Nosé-Hoover thermostat[75,76] was used to maintain the temperature at 310 K with the protein, the membrane, and the solvent (water and KCl) coupled to separate temperature baths with a time constant of 1.0 ps. The Parrinello-Rahman barostat[77,78] was used for semi-isotropic pressure coupling at 1 atm with a time constant of 5 ps and a compressibility value of $4.5 \times 10^{-5}$ bar$^{-1}$.

The first 200 ns of each simulation was discarded (equilibration). Means over time and repeats are reported wherever applicable, where the standard errors were calculated using the standard deviations divided by the square root of number of repeats. Visual Molecular Dynamics (VMD)[79] program was used for visualization, calculation of the 3-dimensional occupancy maps, and the solvent accessible surface area (SASA) calculations. The 3-dimensional occupancy maps were calculated over the last 800 nsec of all repeats after the protein was superposed only along the membrane surface. The Surface Cholesterol Saturation (SCS) was defined as the ratio between the cholesterol- and membrane-exposed surface areas of the protein, both of which were estimated based on SASA. A straight line was fit to log([Chol]) vs $\log\left(\frac{1}{1-SCS}\right)$ data (as in the logarithmic rearrangement of the Hill–Langmuir Equation) to estimate the parameters for the Hill Equations fits to SCS Curves and their associated errors were calculated using bootstrapping with 1000 iterations. The two-dimensional (lateral) radial distribution functions ($g(r)$) of CHOL were calculated using gmx rdf with the protein surface used as a reference and normalized to one at a large distance (1.4 nm). MDTraj[80] was used to superposing the protein and calculating Root Mean Square Deviations (RMSD) for each $C_\alpha$. The clustering of residues to distinct binding sites in Supplementary Fig. 4b was performed on the wild-type protein in 20:80 (mol%:mol%) CHOL:POPC membrane. We used the KMeans algorithm based on a correlation distance matrix $D(i, j) = \sqrt{2(1 - C(i, j))}$, where $C(i, j)$ is the average Pearson correlation coefficient between protein residues i and j, calculated based on the pairwise minimum distance between each protein residue and cholesterol molecules in the system. The number of clusters were determined using the elbow method. The occupancy plots show what portion of the identified clusters are occupied on average over time and independent repeats. The Linear Discriminant Analysis (LDA) was performed using Scikit-learn[81]. The trajectories were first read and superposed on the xy-plane (membrane plane) using the transmembrane portion of the proteins. The coordinates of all common heavy atoms of the WT protein were used as the feature vector. A train-test split of 3:7 was applied to shuffled data. The LDA model for WT M-protein results in perfect prediction score for the test set. In a data set composed of 4 classes (4 different membrane compositions), LDA resulted in at most 3 components by construction. In our case, the first LDA component had an explained variance ratio of 90%, and projection of the feature vector onto the first LDA component was enough to separate all classes.

## Statistical analysis

Statistical analyses were performed using the GraphPad Prism software package (v8.0.0). For graphs with multiple comparisons being made, two-way ANOVA was performed with Dunnett's or Fisher's LSD test for multiple comparisons. For graphs with single comparisons being made, either a ratio paired t test or an ordinary one-way ANOVA with Dunnett's test for multiple comparisons were performed. For colocalization analysis a non-parametric Mann–Whitney or Kruskal–Wallis test were performed. Exact P-values are given within the figure legends. For all experiments, data are reported from biological replicates. No reported data are from repeated measurements of the same samples.

## Reporting summary

Further information on research design is available in the Nature Portfolio Reporting Summary linked to this article.

## Data availability

The source data underlying Figs. 1c–e, 2d, e, 3e, f, 4b, c, e, 5a, c–e, g, 6a–d, f, g, 7a, b, d–h, and Supplementary Figs. 1c, 2b, d, 2a, b, 4a–c, 5b–d, 6a–h, 7a–c, 9a, b, 10b, c, 11b, d are provided in the Source Data file. The MD simulations data generated in this study have been deposited online and can be accessed via doi: 10.5281/

zenodo.8268517. The original protein structure used for MD simulations (PDB ID: 5IRE)[20] has been previously deposited online and can be accessed via https://doi.org/10.2210/pdb5IRE/pdb. The published article includes all datasets generated and analyzed during this study. Source data are provided with this paper.

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

## Acknowledgements

We gratefully acknowledge the Electron Microscopy Core Facility (EMCF) of Heidelberg University, headed by Charlotta Funaya, for expert support and access to their equipment. We acknowledge the Infectious Diseases Imaging Platform (IDIP), headed by Vibor Laketa, at the Center for Integrative Infectious Disease research, Heidelberg, Germany, for access to their equipment. We thank Mathias Mayer at the Center for Molecular Biology Heidelberg (ZMBH) for access to the CLARIOstar plate reader. Furthermore, we thank Alain Kohl, Glasgow, UK, for provision of the mosquito cell line Aag2 and the European Virus Archive (EVAg) for provision of the ZIKV H/PF/2013 strain. We are grateful to Ulrike Herian, Marie Bartenschlager and Micha Fauth for their excellent technical support and Pietro Scaturro, Leibnitz Institute for Virology (LIV) in Hamburg for providing the initial overexpression construct for the ZIKV prM-HA protein. Furthermore, we thank Felix Pahmeier for generating the ZIKV reporter construct. Work by R.B. was supported by grants from the Deutsche Forschungsgemeinschaft (DFG, German Research Foundation) – Project number 112927078 – TRR83 and Project number 240245660 – SFB1129 and by a grant from the Helmholtz Association's Initiative and Networking Fund (Project COVIPA; KA1-Co-02). S.D. was supported by a Humboldt Research Fellowship for Postdoctoral Researchers. V.P. was supported by a European Molecular Biology Organization (EMBO) long-term fellowship (ALTF 454-2020). The Vattulainen group has been supported by the Academy of Finland (projects 331349, 336234, 346135), the Sigrid Juselius Foundation, Helsinki Institute of Life Science (HiLIFE) Fellow Program, the Human Frontier Science Program (RGP0059/2019), and DFG (SFB/TRR 83). We acknowledge the computing resources provided by the CSC – IT Center for Science Ltd. (Espoo, Finland) and LUMI supercomputer, owned by the EuroHPC Joint Undertaking, hosted by CSC and the LUMI Consortium. The Lozach group has been supported by grants from the DFG – grant number LO-2338/3-1 and from Agence Nationale de la Recherche (ANR) – grant numbers ANR-21-CE11-0012, ANR-22-CE15-0034, and ANR-11-IDEX-0007. For the publication fee we acknowledge financial support by Deutsche Forschungsgemeinschaft within the funding program "Open Access Publikationskosten" as well as by Heidelberg University.

## Author contributions

S.G. and R.B. designed the concept of the study. S.G. performed the experiments, analyzed the data, and interpreted the results. G.E., W.K., and I.V. designed the simulation part and aided in writing the manuscript. G.E. and W.K. performed the MD simulations. G.E. analyzed the data of molecular dynamics simulations and helped in writing the manuscript. V.P. conceptualized and performed fusion assays with R18-labeled ZIKV and analyzed the data. S.D. aided with confocal microscopy. S.E. set up the photoaffinity labeling assay using the PAC-cholesterol probe. G.M. aided in data acquisition for EM experiments. L.W. performed parts of the experiments. Z.M.U. performed parts of the experiments. Z.M.U. and P.Y.L. provided expertize on entry assays. T.M.L. aided in data acquisition. U.H. provided expertize and technical assistance for EM sample preparation and acquisition. B.B. provided expertize and technical assistance for biochemical lipid assays. S.G. and R.B. wrote the manuscript. R.B. supervised the study and secured funding. All authors reviewed and approved the manuscript.

## Funding

## Competing interests

The authors declare no competing interests.
