## [Peer Review File · Nature Communications]

Zika virus prM protein contains cholesterol binding motifs required for virus entry and assemblyREVIEWER COMMENTS

Reviewer #1 (Remarks to the Author):

This paper combines variety of experiments and molecular dynamics (MD) simulations to identify cholesterol binding sites in the Zika virus structural protein prM, and delineate the effects of mutants lacking this binding site. My review focuses on the simulation component of this study.

A series of 1 microsecond simulations were carried out on a dimer of the M domain of prM in POPC bilayer at four different cholesterol mole fractions (0, 10, 20, and 30) on the wild type (WT) and 4 mutations in the cholesterol binding site). Ten replicates were generated for each of the 20 systems (4 cholesterol concentrations and 5 protein variants) for a total of 200 microseconds, an exemplary effort by current standards. While the simulation results (presented entirely in Figure 3) are reasonably convincing and are consistent with the experimental data, I have the following questions/comments:

1. The simulation snapshot in Fig 3c of the WT shows tight binding of a cholesterol to the purported cholesterol binding site (CRAC, for Cholesterol Recognition Amino acid Consensus sequence). In contrast the snapshots of the four mutants show cholesterol binding to a non-CRAC site in one, and no cholesterol near any of the other three mutant structures. However, 5 snapshots hardly count as proof of binding specificity, especially given the radial distribution functions for protein cholesterol (Fig 3d), where the different proteins seem more similar than different. Similarly, the Surface Cholesterol Saturation (SCS) plots in Fig 3e, show the highest SCS for WT, but the mutants are not so much lower. From these plots and what the authors state I infer that the binding of cholesterol is specific and strong for the CRAC sequences of the WT, and nonspecific and weak for the mutants. However, the authors need to show this clearly and convincingly. For example, present plots or tables of the number of unique binding events and residence times of cholesterol to assorted binding sites for all of the 20 systems (these can be divided into CRAC and "other". There are only 7 figures in the ms, and the journal allows 10. Perhaps this can be one of the remaining 3, or placed in the SI.

2. I am not convinced that the carrying out 10 replicates of 1 microsecond, is more appropriate than 5 of 2 microseconds, or even 2 of 5. The choice really depends on the longest residence time of cholesterol to specific sites. If no residence is longer than 1 microsecond, what the authors did is fine. Otherwise, the analysis is more complex. The average distance of cholesterol diffusion is also a consideration. I realize that the authors cannot easily extend their simulations at this point, but there should be some justification in the Methods as to why 1 microsecond replicates were chosen.

3. Returning to Fig 3d (the $g(r)$), the plotting style (which seems to include all 10 replicates for each system) obscures the data. I suggest showing only 1 line (the average over the replicates) for each protein type. Also, I note that the WT (blue) never shows the highest $g(r)$ at short distances. I would have thought that tight specific binding should have yielded the highest max in $g(r)$ at the binding distance. Please explain.

Editorial comments.

Though I found the plots cluttered, the writing was clear and effective. A few minor comments:

Abstract, line 8: This line was confusing to me, specifically the phrase "causing SIKV entry". As I understand the paper, loss of prM association reduces virus entry and leads to assembly defects. Sorry if I've misunderstood something. Pls rewrite if I have not.

Page 6, 11 lines from bottom: "Additionally" is better than "Besides"

Page 7, line 5 of paragraph 2: "...mutants reverted" to "... mutants which reverted"

Page 9, 10 lines from bottom: "by-passed" does not require a hyphen.

Reviewer #2 (Remarks to the Author):

Giellner et al. 2023

'Zika virus prM protein contains cholesterol-binding motifs required for virus entry and assembly" by Goellner et al. describes the discovery of two putative cholesterol binding sites in the ZIKV prM protein by photoaffinity labeling using a bifunctional PAC-cholesterol probe, pull-down and Western blot. The authors present evidence supporting their model of the M-domain of prM facilitating the entry and assembly of infectious virus particles via its two cholesterol-binding motifs. There are a series of well-thought-out experiments meticulously performed listed in this manuscript. However, the authors do not conclusively show that cholesterol binding critically modulates the virus entry or exit. The authors have also not examined the other functions of the M protein and its role in virus assembly. The mutations are not significantly affecting virus release based on TCID50 assays, and contradictory results are observed in EM, showing a significantly reduced number of particles. Authors have not established a critical role of cholesterol binding in ZIKV assembly. As cholesterol is present in the lipid bilayer, it is hard to prove that M protein is recruiting cholesterol molecules for some specific function in the virus life cycle. Therefore, the manuscript shows the lack of a mechanism and the functional significance of the predicted domains. Supporting structural studies using the mutant viruses would have strengthened the manuscript.

Major points

Throughout this manuscript, the presence of cholesterol on the lipid bilayer of an enveloped virus budding from the endoplasmic reticulum membrane has to be addressed. With or without prM binding, cholesterol will be present in the virion, similar to the membrane composition of the ER membrane. Another factor to consider is that high resolutions cryoEM structures of immature and mature flaviviruses including ZIKV, have not identified cholesterol in binding to the prM/M and E proteins whereas several other specific lipid molecules have been identified binding to different pockets. These structural aspects are not addressed in the manuscript.

Cholesterol binding to M protein isolated from the whole cell lysates might not indicate cholesterol interaction in the assembled virions. Could this be done using purified virus produced from the labeled cells?

Since prM expression alone can lead to nonfunctional proteins, as it does not transit to the correct organelle, the experiments of expressing only prM is not similar to what is acceptable as happening in the virus. Authors could have used a VLP construct that would facilitate functional prM/E spike forming particles. Is an HA-tagged prM functional? Does this affect the normal prM protein traffic?

It is unclear how M domains form dimers and why this was used for simulations. M protein forms heterodimers with E in the mature virus, and thus far, no M/M interactions have been reported. Specifically, reference 18 does not report the formation of M dimers. This needs to be clarified. Figure 2A shows completely buried TM helices and perimembrane helices. Is the composition of PM lipid bilayer or ER lipid bilayer used for simulation as they have different thickness? The M protein perimembrane helix is exposed on the luminal side, and the linker sequence is exposed to the cytoplasmic side. The image does not show the correct organization of helices, as seen in the cryoEM structures of the mature virus.

Since prM is involved in a significant interaction with E during virus maturation, the mutational analysis does not show that the reduction in virus titer is due to cholesterol binding.

During flavivirus assembly, prM is known to interact strongly with the E protein. The Western blot in Fig 1 shows a very light band for the Envelope protein, suggesting little to no interaction of prM to E protein pulled down with cholesterol. The authors should provide an explanation.

Cryo-EM structures of ZIKV purified from mammalian cells show the presence of lipid molecules close to prM-E membrane helices. The authors have not discussed this aspect in the manuscript. A comparative analysis of the possible cholesterol binding sites described by the authors in this manuscript and the lipids present in the cryo-EM structure should be included in the discussion.

Line 100: How does data from CARC3 mutant fit in this model?

Line 137: check the amino acid numbering: aa 38-44 of ZIKV.

Lines 293-300; ZIKV spread and CARC2 with the cholesterol solubilizing drug:
After how many days/hr was the virus spread analyzed?

Line 328: The authors say, "CARC3 mutants were severely affected independent of cellular cholesterol levels". Does this mean that CARC3 is not a bonafide cholesterol-binding site? Results show the CARC3 mutations and the resulting loss of C-E colocalization might result from a structural change to M protein rather than any effect of cholesterol binding. Text in the results and discussion are confusing regarding the roles of these two sites and cholesterol.

Line 279: The statement that "The results showed that within the first 48 hours when no virus spread occurs, and only viral replication in transfected cells is monitored" is not an accurate statement based on the published growth kinetics of ZIKV.

Fig 5b. IFA is not of good quality, which makes the data unreliable. The scale bar is 10 μ m. How does this scale bar compare to that in Fig 4d, where the scale bar is 20 μ m?

Cell-to-cell spread of ZIKV can be demonstrated with figures of higher magnification which will also improve the quality and show cell boundaries and nuclei in better resolution. Both Figs 5b and 4d have no DAPI signal. The authors should improve both figures to make the data more reliable.

Fig 5f is not convincing enough to conclude that CARC2 has a role in virus entry. The data show the drug affects all three viruses similarly. Authors should provide quantitative data to prove their point. Why were CRAC3 mutants not included in this experiment?

Minor points

Line 67, Line 77: Please provide references for ZIKV NC.

Line 73: Is it clear whether precursor capsid is associated with LD as LDs are made of monolayers and precursor C has a TM domain?

Line 272: It is not clear how the authors can state that the binding assays show that E protein conformation is intact and not affected by the mutations in the M domain of prM.

Line 279: Is it true that ZIKV does not release virus in 48 hours? It will be useful to provide a standard growth kinetic analysis of WT and mutant viruses including 0-120 hours with 24 h intervals, as majority of the conclusions are based on virus release.

Line 300: Based on the mechanism of flavivirus entry, it is not clear how cholesterol binding of M protein affects virus entry as M is not exposed during entry, and access is entirely dependent on E protein.

Extended Data Figure 2A shows more virus is released from the mutants at early time points. It is important to show growth kinetic assays, as 120 hours seems too late to have cell viability. Does the specific infectivity change over time?

Line 357: Since the virus release is not significantly reduced in the mutant virus, how is the reduction in virus particles calculated by EM analysis explained?

Please provide a catalog number for PAC-cholesterol purchased from Sigma Aldrich.

Reviewer #3 (Remarks to the Author):

The manuscript from Goellner et al. describes the identification of two functional cholesterol binding motifs within the transmembrane domain of the structural protein prM of Zika virus. The sequences are well conserved across the flaviviruses and suggest that these binding motifs are important for all the flaviviruses. Using photoaffinity labeling, molecular dynamic simulations, reverse genetics and virology assays, the authors present a convincing set of data that cholesterol binding is real and fulfills a functional requirement for entry and virus assembly. Interestingly, the mutational data suggests the CARC2 site functions distinctly from the CARC3 site. The manuscript is an important contribution to the field. The data are strong, and the text is well written.

Specific comments:

Line 133: "We identified..." The sentence sounds awkward and should be re-written.

Line 141: Did the authors introduce substitutions into both CARC2 and CARC3 sites together? If not, why? Is it possible given their positions (see Fig. 3a) that CARC1 and CARC2 collectively contribute to binding?

Figure 3: I am confused by the modeling simulations shown in panel c. Is there only one cholesterol binding site? Are sites 2 and 3 physically distinct?

Line 430: Given the propensity of data suggesting cholesterol binding, have high resolution cryo-EM structures found cholesterol in the predicted binding motifs?

REVIEWER COMMENTS

Reviewer #1 (Remarks to the Author):

This paper combines variety of experiments and molecular dynamics (MD) simulations to identify cholesterol binding sites in the Zika virus structural protein prM, and delineate the effects of mutants lacking this binding site. My review focuses on the simulation component of this study.

A series of 1 microsecond simulations were carried out on a dimer of the M domain of prM in POPC bilayer at four different cholesterol mole fractions (0, 10, 20, and 30) on the wild type (WT) and 4 mutations in the cholesterol binding site). Ten replicates were generated for each of the 20 systems (4 cholesterol concentrations and 5 protein variants) for a total of 200 microseconds, an exemplary effort by current standards. While the simulation results (presented entirely in Figure 3) are reasonably convincing and are consistent with the experimental data, I have the following questions/comments:

1. The simulation snapshot in Fig 3c of the WT shows tight binding of a cholesterol to the purported cholesterol binding site (CRAC, for Cholesterol Recognition Amino acid Consensus sequence). In contrast the snapshots of the four mutants show cholesterol binding to a non-CRAC site in one, and no cholesterol near any of the other three mutant structures. However, 5 snapshots hardly count as proof of binding specificity, especially given the radial distribution functions for protein cholesterol (Fig 3d), where the different proteins seem more similar than different.

[Response] We thank the reviewer for giving us the opportunity to clarify the text and Figure caption of Fig. 3c. The reviewer is correct that 5 snapshots do not count as proof of binding. However, the images in Fig. 3c are not mere snapshots, they are iso-occupancy surfaces averaged over the last 800 ns of 10 simulation repeats (each 1 microsecond long). Thus, the results shown in Fig 3c is based on 8 microseconds of averaged simulation data. For these calculations, we concatenate all 10 800-ns-long trajectories separately for each mutant. We superpose the trajectories based on the initial structure only on the xy-plane (membrane plane). We use these superposed trajectories to calculate the volumetric occupancy maps. We, then, choose an iso-occupancy value that shows specific localization of cholesterol. At 16% iso-occupancy value we can see well-defined cholesterol density near the protein. If we increase the iso-occupancy value, then the density diminishes as expected, and at smaller values, it covers the whole membrane. Concluding, we observe specific localization of cholesterol near the protein surface, but it is only observed in the wild-type protein.

We also agree with the reviewer that the radial distribution functions as such do not reveal major differences. The point we tried to make with the RDF plots was that cholesterol density is enhanced at similar levels near the M-protein dimer regardless of the mutation or the membrane cholesterol concentration. However, when taken together with the iso-occupancy maps, one can see that only in the wild-type system, cholesterol binds a specific region on the protein surface.

Similarly, the Surface Cholesterol Saturation (SCS) plots in Fig 3e, show the highest SCS for WT, but the mutants are not so much lower.

[Response] The referee is correct. Our data also suggest that there may be some cholesterol levels where these effects are pronounced. There may be not enough cholesterol association at low cholesterol concentrations (10 mol%) even to the wild-type and at high cholesterol levels, there may be some mutants that show similar level of SCS. We note that SCS does not measure any specific binding, but we infer the level of cholesterol coverage of the M-protein surface. This analysis reveals more subtle differences that RDFs cannot capture.

From these plots and what the authors state I infer that the binding of cholesterol is specific and strong for the CRAC sequences of the WT, and nonspecific and weak for the mutants.

[Response] Our results show that M-protein favors interactions with cholesterol. The mutations reduce these effects. It also shows that wildtype M-protein can bind cholesterol specifically, but this depends on the cholesterol level. We only observed the localized cholesterol interactions in 20 mol%.

However, the authors need to show this clearly and convincingly. For example, present plots or tables of the number of unique binding events and residence times of cholesterol to assorted binding sites for all of the 20 systems (these can be divided into CRAC and “other”. There are only 7 figures I the ms, and the journal allows 10. Perhaps this can be one of the remaining 3 or placed in the SI.

[Response] We clarified this point in the text. Our results do not strongly suggest a direct interaction with the CRAC domains. Instead, the data show that the mutations in the putative CRAC domains reduce the surface coverage of the protein by cholesterol and diminishes specific localization of cholesterol in the wild-type system.

We have performed additional analysis to detect the cholesterol binding sites based on the data. For this purpose, we clustered the protein residues based on the correlations of the cholesterol binding. This is a way to identify distinct binding sites. The residues form three major groups, each corresponding to one of the CARC domains (Supplementary Figure 4a). We also estimated from the simulations how much each site is occupied by cholesterol (Supplementary Figure 4a). Our results show that Site 1, that is formed by the perimembrane helices do not substantially interact with cholesterol as expected. Sites 2 and 3 have similar levels of occupancy in the wild-type in 20:80 CHOL:POPC membrane. All mutations cause a decrease in the occupancy of Sites 2 and 3.

2. I am not convinced that the carrying out 10 replicates of 1 microsecond, is more appropriate than 5 of 2 microseconds, or even 2 of 5. The choice really depends on the longest residence time of cholesterol to specific sites. If no residence is longer than 1 microsecond, what the authors did is fine. Otherwise, the analysis is more complex. The average distance of cholesterol diffusion is also a consideration. I realize that the authors cannot easily extend their simulations at this point, but there should be some justification in the Methods as to why 1 microsecond replicates were chosen.

[Response] We also would refrain from making such a comment as “10 replicates of 1 microsecond, is more appropriate than 5 of 2 microseconds, or even 2 of 5”. The aim of having many replicates is to randomize the initial conditions for better sampling. We performed the

residence time analysis as suggested by the referee and the results show that no residence time is greater than 1 microsecond (Rebuttal Fig. 1). Only in the 20:80 CHOL:POPC system, we see a significant difference in the residence time of cholesterol binding to the protein surface. And in all cases the residence times are much lower than 1 μ s. Indeed, the maximum residence time (when calculated for each repeat and simulation) is lower than 300 ns. This analysis is also in line with the analysis we presented in the manuscript (Supplementary Fig. 4c).

Rebuttal Fig. 1. The cholesterol residence time distributions. The distributions are generated by bootstrapping survival curves, to each of which a double exponential is fitted. Survival curves are generated using a cutoff of minimum distance between cholesterol and the protein of 0.6 nm.

To confirm that our findings are not affected by the length of the simulations, we carried out additional simulations. We extended the wild-type M-dimer simulations to 2 μ sec to see if the effects change. Cholesterol binding to the protein shows similar features and remains the same in the second half of the 2 μ sec simulation (Rebuttal Fig. 2) as in the first half (Fig. 3).

Rebuttal Fig. 2. Cholesterol iso-occupancy surface calculated from 1.2-2.0 μ sec of the wild-type M-protein in the 20:80 (mol%:mol%) CHOL:POPC membrane.

3. Returning to Fig 3d (the $g(r)$), the plotting style (which seems to include all 10 replicates for each system) obscures the data. I suggest showing only 1 line (the average over the replicates) for each protein type. Also, I note that the WT (blue) never shows the highest $g(r)$ at short distances. I would have thought that tight specific binding should have yielded the highest max in $g(r)$ at the binding distance. Please explain.

Figure 3d does not include all 10 replicates but the line thickness indicates the error of the $g(r)$. For this purpose, we calculated the $g(r)$ for each system and represented the data and mean and standard error of the $g(r)$. The purpose of the plot was indeed to show that the $g(r)$ does not significantly differ among systems but to highlight all variants of the protein interact with cholesterol, i.e., the cholesterol distribution is enhanced on the surface of the protein.

We believe that $g(r)$ does not contribute to the story significantly enough and may cause confusion. We have therefore moved these data to the Supplementary Information.

Editorial comments.

Though I found the plots cluttered, the writing was clear and effective. A few minor comments:

Abstract, line 8: This line was confusing to me, specifically the phrase “causing ZIKV entry”. As I understand the paper, loss of prM association reduces virus entry and leads to assembly defects. Sorry if I’ve misunderstood something. Pls rewrite if I have not.

[Response] We are sorry for the confusion. We have now modified the sentence as suggested and hope it eases the reading.

Page 6, 11 lines from bottom: “Additionally” is better than “Besides”

[Response] The word has now been exchanged as suggested.

Page 7, line 5 of paragraph 2: “...mutants reverted” to “... mutants which reverted”

[Response] The sentence has now been modified and we hope it eases the reading and improves the understanding.

Page 9, 10 lines from bottom: “by-passed” does not require a hyphen.

[Response] The hyphen has now been removed.

Reviewer #2 (Remarks to the Author):

Goellner et al.. 2023

‘Zika virus prM protein contains cholesterol-binding motifs required for virus entry and assembly’ by Goellner et al.. describes the discovery of two putative cholesterol binding sites in the ZIKV prM protein by photoaffinity labeling using a bifunctional PAC-cholesterol probe, pull-down and Western blot. The authors present evidence supporting their model of the M-domain of prM facilitating the entry and assembly of infectious virus particles via its two cholesterol-binding motifs. There are a series of well-thought-out experiments meticulously performed listed in this manuscript. However, the authors do not conclusively show that cholesterol binding critically modulates the virus entry or exit.

[Response] We thank the Reviewer for giving us the opportunity to clarify the text. We would like to point out that we have followed up on our ZIKV mutants (CARC2), which exhibit entry defects. We have generated a new set of data providing evidence that cholesterol sensing by the M-domain is a determinant for virus fusion. ZIKV CARC2 mutants were found to prematurely fuse in proximity to the plasma membrane, which resembles the phenotype observed for cells that have been cholesterol depleted by treatment with M β CD prior to virus infection. This data has been added to Fig. 5 and Supplementary Fig. 7. Thus, apart from the E protein also the M protein plays a role in virus entry.

The Reviewer is right that we did not show much data on virus egress. However, we would like to point out that the ZIKV mutants (CARC3) are defective in virus assembly, and therefore, an additional egress phenotype is almost impossible to be addressed. However, since this Reviewer suggested to use a ZIKV VLP expression system for the PAC-cholesterol assay, we have used this system as well to further study whether VLP formation and subsequent secretion is affected by the mutations targeting the CARC3 motif. The data has now been added as a new subpanel to Fig. 6. Using the ZIKV VLP system we could show that secretion is reduced for the CARC3 mutants when compared to the secretion of WT VLPs. We have included this aspect in the text.

The authors have also not examined the other functions of the M protein and its role in virus assembly.

[Response] We apologize if we have missed some functions of the M protein and its role in virus assembly. To this end, we have addressed reported functions of the prM protein as follows:

- a. prM and virus maturation (Pierson & Diamond (2012), doi:10.1016/j.coviro.2012.02.011): prM cleavage into the soluble pr peptide and the membrane bound M-domain was observed by western blot analysis (Supplementary Fig. 3 and Supplementary Fig. 10).
- b. prM and its interaction with the E protein (Li et al. (2008), doi: 10.1126/science.1153263): The interaction between prM and Envelope is required to facilitate budding of newly assembled virions into the ER lumen. First, to ensure that our mutations do not affect sub-cellular localization to the ER, we have performed immunofluorescence microscopy staining both the prM protein and reticulon-3, a well-established marker for the ER (Supplementary Fig. 12). Second, to ensure that the interaction between prM and E still occurs, we have performed co-immunoprecipitations (Fig. 7) and immunofluorescence microscopy to study the co-localization of prM and E proteins (Supplementary Fig. 12). To strengthen our study, we have now added the secretion of ZIKV VLPs (Fig. 6d) and have quantified the co-occurrence coefficients (M1 and M2) for the data in Supplementary Fig. 12. Please note that for the CARC3 mutants and WT similar values were observed.
- c. prM protects E proteins and prevents premature fusion (Nambala & Su (2018), doi: 10.3389/fmicb.2018.01797): to rule out that pre-mature fusion occurred we have quantified titers of infectious extracellular virus via TCID50 assays. Our data shows that CARC2 mutants are not significantly different when compared to WT whereas for CARC3 mutants a significant reduction was observed for both extra- and intracellular titers of infectious virus. Furthermore, using a CLEM approach we have found that CARC3 mutants are impaired in their efficacy to assemble new progeny virions, thus further downstream defects

such as egress or fusion within the TGN is technical not feasible to be addressed and even if it would occur, play a minor role.

- d. The M-domain forms a viroporin (Brown et al. (2021), doi: [10.1101/2021.03.11.435022](https://doi.org/10.1101/2021.03.11.435022)): This proposed function is not convincing as the referenced study used overexpressed M proteins alone without the pr peptide and the E protein. Observed oligomerization can therefore be an artefact instead of representing the true nature in a virus particle. Furthermore, such structures (hexamers) should have been identified with the high resolution cryo-EM data, but these have not been detected.
- e. Last, our manuscript provides the novelty by revealing a function of the M domain of the prM protein in virus assembly based on its ability to associate with cholesterol. Thus, we kindly ask Reviewer 2 to provide us with the respective literature on which the claim is founded that we have missed, and therefore did not address, all functions of the M protein in the viral life cycle/ virus assembly.

The mutations are not significantly affecting virus release based on TCID50 assays, and contradictory results are observed in EM, showing a significantly reduced number of particles.

[Response] We apologize that the nomenclature of the ZIKV mutants led to confusion. We have adapted the manuscript in which the mutants are referred to as 2-A/S or 3-A/S. We hope that this eases the reading and prevents confusion. However, the data Reviewer 2 refers to here originated from the two different CARC mutants. It is correct that the TCID50 assays do not show a reduction for CARC₂ mutants, but for EM we did not use the CARC₂ mutants. Instead, for EM analysis we used the CARC₃ mutants which clearly show a significant reduction in TCID50 assays (Fig. 6).

Authors have not established a critical role of cholesterol binding in ZIKV assembly. As cholesterol is present in the lipid bilayer, it is hard to prove that M protein is recruiting cholesterol molecules for some specific function in the virus life cycle. Therefore, the manuscript shows the lack of a mechanism and the functional significance of the predicted domains. Supporting structural studies using the mutant viruses would have strengthened the manuscript.

[Response] We thank Reviewer 2 for this comment and the opportunity to broaden our study. In fact, we have performed experiments using the mosquito cell line Aag2, in which we have characterized ZIKV growth kinetics in media supplemented either with delipidated FBS or normal lipidated FBS. Our results show that cholesterol is not always needed as ZIKV growth kinetics are comparable between the two tested conditions. As mosquito cells are lacking enzymes required for the *de novo* synthesis of cholesterol one can be certain that ZIKV grown in DL-FBS and mosquito cells does not incorporate cholesterol. However, if cholesterol is present, it can get incorporated. These new data have been added to the manuscript as Supplementary Fig. 8. The recruitment of cholesterol by the prM protein is one possible scenario which we have proposed as model/mechanism. Alternatively, cholesterol can be enriched in microdomains in the ER membrane prior to infection and as virus infection progresses, virus assembly can occur at such microdomains. Which of both scenarios occurs is experimentally and technically not possible to address. Thus, we have rephrased our model to ease reading and avoid confusion.

Major points

Throughout this manuscript, the presence of cholesterol on the lipid bilayer of an enveloped virus budding from the endoplasmic reticulum membrane has to be addressed. With or without prM binding, cholesterol will be present in the virion, similar to the membrane composition of the ER membrane.

[Response] We thank the Reviewer for this comment and to provide us with the opportunity to broaden our study. As mentioned above, we could show that cholesterol is not always present in the virion, especially when viruses are produced in the vector host – the mosquito. However, interference with the cholesterol biosynthesis pathway (e.g. treatment with statins, España et al. (2019), doi: 10.1038/s41598-019-47956-1) impairs virus production in the mammalian host. Both findings support the idea that cholesterol is required in the mammalian host. However, we do not agree with the statement that the membrane composition of the virus is similar to the ER membrane. This statement implies that upon infection intracellular membranes remain unaltered with respect to their lipid composition. To this end no comparative analysis has been performed in which the lipidome of the ER was compared to the lipidome of flaviviruses. Instead, the lipidomes of West Nile virus (Martin-Acebes et al., 2014, doi: <https://doi.org/10.1128/jvi.02061-14>) and dengue virus (Hitakarun et al., 2022, doi: <https://doi.org/10.3390/v14112566>) have been published alongside to the lipidomic changes of cell pools infected with the respected virus. Apart from this, several publications have highlighted that the virus lipid envelope is distinct from the membrane where it originates from (e.g. HCV (Merz et al., 2011, doi: <https://doi.org/10.1074/jbc.M110.175018>), HIV (Brügger et al., 2006, doi: <https://doi.org/10.1073/pnas.0511136103>), Semliki Forest virus (Renkonen et al., 1971, doi: [https://doi.org/10.1016/0042-6822\(71\)90033-X](https://doi.org/10.1016/0042-6822(71)90033-X)), SARS-CoV2 (Saud et al., 2022, doi: 10.1016/j.jlr.2022.100208), bovine viral diarrhea virus (Callens et al., 2016, doi: <https://doi.org/10.1371/journal.ppat.1005476>), Influenza virus (Gerl et al., 2012, doi: 10.1083/jcb.201108175)). Furthermore, we would like to point out that the ER, from where ZIKV obtains its lipid envelope, is low in cholesterol.

To answer the question whether the lipid composition of ZIKV resembles the lipid composition of the ER, would have exceeded the scope of the current study but will be addressed in the future.

Another factor to consider is that high resolutions cryoEM structures of immature and mature flaviviruses including ZIKV, have not identified cholesterol in binding to the prM/M and E proteins whereas several other specific lipid molecules have been identified binding to different pockets. These structural aspects are not addressed in the manuscript.

[Response] We apologize to have missed this point in our discussion. We have modified our manuscript and have taken this aspect into account.

Reviewer 2 is correct with the statement that cryo-EM structures have analyzed immature and mature virus particles belonging to the flavivirus genus in which lipids have been identified. However, we would like to point out the following reasons why cholesterol could not have been identified by cryo-EM so far:

- a. Identification of lipids and lipid pockets in Spondweni virus (Renner et al. 2021, <https://doi.org/10.1038/s41467-021-21505-9>), dengue and West Nile virus (Hardy et al. 2021, <https://doi.org/10.1038/s41467-021-22773-1>): These structural analyses were per-

- formed on virus particles derived from mosquito cells, which are lacking enzymes essential for *de novo* synthesis of cholesterol and lipidomic profiles of mosquito cells have not identified cholesterol, thus its detection in virions is not possible.
- b. Apart from these reports structural analysis was done on ZIKV (Sirohi et al., 2016, doi: 10.1126/science.aaf5316; Seevana et al., 2018, doi: 10.1016/j.str.2018.05.006), yet these reports were solely focusing on the structural organization of the particles and do not provide information towards lipids.
 - c. The literature we assume Reviewer 2 is referring to could be from DiNunno and colleagues (2020, doi: 10.1038/s41467-020-18747-4), which in fact analyzed the lipid pockets found in ZIKV particles recovered from mammalian cells. The authors described a differential lipid pocket that arises in virions derived from mammalian cells, which is absent in virions derived from mosquito cells. However, the authors do not make a claim of a specific lipid that is incorporated but instead state that it appears to be a crooked lipid.
 - d. To this end, no specific lipid was shown to get incorporated but rather modeling and data obtained from lipidomic profiling of virions were used to model either PE (Renner et al., 2021) or phosphatidylcholine (PC) (Pulkkinen et al., 2022, doi: 10.3390/v14040792) as these are the most common lipid species found in virions. To resolve lipids by cryo-EM it is of utmost importance that within each sample the lipid is fixed and abundant in a single position to allow its reconstruction. To this end we cannot say how many cholesterol molecules are incorporated into one virus particle and whether its incorporation varies in terms of position in the virion itself. Slight changes in position will prevent its identification via cryo-EM. In fact, our MD simulation data show that cholesterol binding is quite dynamic and thus may cause difficulties in specific capturing in cryo-EM if cholesterol is not cross-linked to proteins in proximity.

Cholesterol binding to M protein isolated from the whole cell lysates might not indicate cholesterol interaction in the assembled virions. Could this be done using purified virus produced from the labeled cells?

[Response] We thank Reviewer 2 for this comment and for giving us the opportunity to broaden our study. Although we did not perform photoaffinity pull down assays on purified viruses produced in labelled cells, we instead used a visualization approach of incorporated PAC-cholesterol into virus-like particles. For this we transfected Huh7-Lunet T7 cells with the ZIKV VLP expression construct and 18 hours post transfection labelled the cells with PAC-cholesterol and allowed VLP production in the presence of the lipid probe for 4 hours. Thereafter, we collected the supernatants and purified and washed the ZIKV VLPs using a 100K Amicon filtration device (this guarantees the removal of free PAC-cholesterol). Thereafter, VLPs were exposed to UV light to facilitate cross-linking of our lipid probe to adjacent proteins. Thereafter, we click labelled our probe to an azide fluorescent dye and subjected our samples to western blot analysis. PAC-cholesterol was visualized in the gel using a fluorescent imager, whereas viral proteins were detected by chemiluminescence upon transfer onto a PVDF membrane. Our result (Fig. 1f) shows that three distinct bands appeared in supernatants containing ZIKV VLP, which were absent in supernatants of mock transfected cells. Superimposition of the Cy5.5 signal with the western blot image demonstrates that we do have a co-occurrence of the cholesterol probe with the viral proteins E, prM and M. Notably, the strongest signal of the cholesterol probe was found for the M protein. Thus, we can conclude that cholesterol is incorporated into the ZIKV VLP (and most likely virion) envelope and there, cholesterol is closely associated with the M-domain of prM.

Since prM expression alone can lead to nonfunctional proteins, as it does not transit to the correct organelle, the experiments of expressing only prM is not similar to what is acceptable as happening in the virus. Authors could have used a VLP construct that would facilitate functional prM/E spike forming particles. Is an HA-tagged prM functional? Does this affect the normal prM protein traffic?

[Response] We thank Reviewer 2 for this comment and providing us the opportunity to improve the status of our manuscript. We now have characterized the prM-HA construct by performing immunofluorescence microscopy assay to determine the subcellular localization of WT and CRAC/CARC mutants. As positive control we have analyzed ZIKV infected cells in parallel. Our results show that the subcellular localization (ER stained by RTN3) is not affected and well comparable for prM-HA constructs as observed for infected cells. These data have now been added to the manuscript as Supplementary Fig. 1.

To address the function whether prM-HA can interact with E to form subviral particles we have collected supernatants of co-transfected cells and therein contained proteins were purified and analyzed via western blot to validate the secretion of E and prM-HA proteins. The result shows that co-transfected cells produce and release ZIKV VLPs (single proteins would have been washed away as we have used a 100K AMICON filtration device). Furthermore, western blot analysis showed two bands for HA-probed proteins, thus suggesting a secretion route via the *trans* Golgi where the furin protease cleaves prM to give rise to the soluble pr peptide and the membrane anchored M protein. This data has now been added to Supplementary Fig. 2 in the manuscript. In addition to these two assays, we also have investigated the colocalization of prM and E by performing immunofluorescence assay. Both proteins shared similar co-occurrence coefficients.

Although we could show that the HA-tagged prM protein is fully functional, we have repeated our PAC-cholesterol assay using the ZIKV VLP system in which we did observe a similar defect of the CARC2 and CARC3 mutants with respect to their binding ability to cholesterol. This data has now been added as Supplementary Fig. 3 to the manuscript.

It is unclear how M domains form dimers and why this was used for simulations. M protein forms heterodimers with E in the mature virus, and thus far, no M/M interactions have been reported. Specifically, reference 18 does not report the formation of M dimers. This needs to be clarified.

[Response] It is correct that to this end no biochemical assays have been performed proving evidence that the M protein forms homodimers. Furthermore, we know that the transmembrane domains of the prM protein forms heterodimers with the E protein. However, to this end many studies have depicted the E/M heterodimer in a dimeric form that has the E/M–M/E form. The homodimer of M is sandwiched between the transmembrane domains of two E proteins. This can be observed in the single unit of the Cryo-EM structure. Indeed, Figure 1D in Sevana et al.. (2018) (<https://doi.org/10.1016/j.str.2018.05.006>) shows this organization clearly. However, we have replaced this reference with Sirohi et al., 2016 (<https://www.science.org/doi/10.1126/science.aaf5316>), which we have used for our MD simulations and a similar organization is depicted in Figure 2B. Apart from these two reports others have used a similar representation of two heterodimers (Zhang et al., 2013,

doi: [10.1038/nsemb.2463](https://doi.org/10.1038/nsemb.2463); Nicholls et al., 2020 doi: [10.1016/bs.aivir.2020.08.003](https://doi.org/10.1016/bs.aivir.2020.08.003); Sirohi et al., 2016, doi: [10.1126/science.aaf5316](https://doi.org/10.1126/science.aaf5316); Sevvana et al., 2018 doi: [10.1016/j.str.2018.05.006](https://doi.org/10.1016/j.str.2018.05.006)).

We used only the M protein for this study for two main reasons: 1) The experiments have shown no cholesterol interaction with the E proteins; 2) and to get sufficient MD sampling of the protein-cholesterol interactions by reducing the system size.

We would like to point out that a recent study by Pulkkinen and colleagues (2022, doi: [10.20944/preprints202204.0001.v1](https://doi.org/10.20944/preprints202204.0001.v1)) did find an interaction between M proteins in purified virus particles for TBEV.

We also conducted additional simulations of the E/M–M/E dimer in a 20:80 Chol:POPC bilayer (10 repeats x 2000 ns), aimed at verifying cholesterol binding to the M-protein in the context of E-M-M-E. Due to the larger system sizes, obtaining sufficient sampling proved to be more challenging. To address this, we extended the simulation time by a factor of two.

Analyzing the results, we identified cholesterol binding spots in iso-occupancy maps. Notably, the data revealed a significantly high occupancy cholesterol binding site near one chain of the M-protein (Rebuttal Fig. 3). Additionally, we observed another site with lower occupancy near the E-protein (Rebuttal Fig. 3). These findings support our previous results about cholesterol binding to the M-protein, but do not rule out cholesterol–E-protein interactions. Although our experiments show no cholesterol binding to E-proteins, further control simulations would be necessary to confirm this *in silico*, but such studies are beyond the scope of our study reported here.

Rebuttal Fig. 3. E/M–M/E simulated in a 20:80 (mol%:mol%) CHOL:POPC bilayer. Two high occupancy sites for cholesterol were identified in 10 simulations: 1) dark green (occupancy = 0.35) and 2) transparent green (occupancy = 0.15).

We have adapted our manuscript, to make it clearer that the M-dimer does not refer to direct protein-protein interactions but was extracted from the two heterodimers as reported by Sirohi et al., 2018.

Figure 2A shows completely buried TM helices and perimembrane helices.

[Response] We replaced Fig. 2A to provide a clearer representation. The transmembrane domain of the M-protein is much shorter than that required to span bilayers. In the representation, the perimembrane helices are indeed in the head group region of the protein and the transmembrane domains extend just pass the center of the bilayer. This placement is based on the positioning of proteins in membranes (PPM) web server. The same positioning is obtained based on the E/M-dimer or M-dimer alone. Furthermore, we have adapted the schematic (cartoon) in Fig. 2A to avoid confusions.

Is the composition of PM lipid bilayer or ER lipid bilayer used for simulation as they have different thickness?

[Response] We used a simpler binary mixture of POPC and cholesterol for the simulations, so the model membranes represent neither PM lipid bilayer nor ER lipid bilayer. The main reason for this choice is to sample the interaction of cholesterol with protein: in the POPC-Chol system, this sampling is more efficient than in multicomponent systems. However, the low cholesterol concentrations are more relevant to the ER, and higher ones are more relevant to PM. PM and ER are 3.6-4.2 nm and 3.8-4 nm, respectively (bionumbers.hms.harvard.edu).

The M protein perimembrane helix is exposed on the luminal side, and the linker sequence is exposed to the cytoplasmic side.

[Response] The perimembrane helix is indeed exposed to the luminal side, but the linker sequence remains within the lipid bilayer. This is determined by the structure of the M-protein. The M-protein would need to adopt a different conformation than those in the cryo-EM structure if it were to span the membrane.

The image does not show the correct organization of helices, as seen in the cryo-EM structures of the mature virus.

[Response] The image shows the correct orientation of the helices, but the referee's comment suggests that we did not originally provide the right context and good representation. Therefore, in Rebuttal Fig. 4 we provide additional explanation addressing the reviewer's comment and to show that the orientation of helices is correct.

Rebuttal Fig. 4. M-protein localization and interaction within the ZIKV membrane. (A) Representation of a single unit of the ZIKV (PDB ID: 5IRE) E/M–M/E complex, showing 2 E-proteins (gray) flanking 2 M-proteins (red and blue). The PPM web server predicts the region between the white lines to be inserted into the membrane. (B) The M-protein dimer (M–M) used for simulations reported in the manuscript. The dimer is extracted from the E/M–M/E complex. The white lines indicate the membrane-inserted region. Both E/M–M/E (A) and M–M (B) show consistent insertion and orientation in the membrane. (C) M–M embedded in a 20:80 Chol:POPC membrane. Green spheres highlight phosphorus atoms in the POPC headgroups. The membrane thickness is approximately 4.4 nm thick based on the phosphorus atoms. The lower leaflet is thinner near the protein due to hydrophobic mismatch. (D) Surface representation of the lipid acyl chains indicating the hydrophobic core of the membrane. (E) Surface representation of the bilayer. Perimembrane helices are exposed to the luminal side, while they lie parallel to the membrane plane at the headgroup region of the POPC molecules.

The perimembrane helices are located within the P-atom region in membranes with lipid compositions of 0:100, 10:90, and 20:80 Chol-POPC. In the case of 30:70 Chol-POPC, the membrane appears significantly thicker, causing the perimembrane helices to extend beyond the P-atom region (Rebuttal Fig. 5).

Rebuttal Fig. 5. Iso-occupancy surfaces representing the 3D number density of P-atoms averaged over trajectories.

To further investigate the M–M insertion into the membrane and ensure accurate positioning of the perimembrane helix, additional simulations were conducted with the protein being less deeply inserted initially. Rebuttal Fig. 6A depicts the initial state. These simulations

were repeated ten times, each running for 1000 ns. In 5 out of 10 simulations (Rebuttal Fig. 6B), the protein achieved full insertion into the membrane, with the perimembrane helices occupying the P-atom region as expected. However, in the remaining 5 simulations, the protein was either expelled from the membrane or the dimer was disrupted.

Rebuttal Fig. 6. M-protein insertion into the membrane.

(A) The membrane is constructed around the M-protein, exposing the perimembrane helices entirely to the surrounding solution. In this configuration, the transmembrane region of the protein spans only one leaflet of the membrane. The simulations demonstrate two possible outcomes after 1000 ns, where the M-protein integrates completely into the membrane (B) or is expelled out of the membrane (C) during the course of the simulation.

Since prM is involved in a significant interaction with E during virus maturation, the mutational analysis does not show that the reduction in virus titer is due to cholesterol binding.

[Response] To ensure that the mutations introduced into the M domain of the prM protein do not affect the interaction with Envelope we have performed co-immunoprecipitation assays (Fig. 7) observing no significant changes for the mutants when compared to WT. Furthermore, we have quantified the co-occurrence coefficients (M1 and M2) for the CARC3 mutants that exhibit the assembly defects. This data has now been added to the Supplementary Fig. 12. We would like to point out that no differences were detected when comparing WT and CARC3 mutants.

We did try to rescue our mutants by performing an experiment in which cholesterol (Sigma Aldrich: C4951-30MG) was added to a final concentration of 50 μ M. However, instead of observing an increase in ZIKV particle production we observed complete abrogation of the ZIKV life cycle (Rebuttal Fig. 7). This could be due to many reasons of which one could be the negative feedback loop induced in cholesterol biosynthesis, thus shutting down the cholesterol production. Whether other steps in the viral life cycle depend on cholesterol needs to be addressed in future work.

Rebuttal Fig. 7. Addition of cholesterol inhibits ZIKV particle production. Huh7-Lunet cells were transfected with synZIKV-H/PF/2013 and treated with 50 μM water soluble cholesterol. At 48 and 72 h.p.t. supernatants were collected, and extracellular infectious titers were estimated via TCID50 assay.

During flavivirus assembly, prM is known to interact strongly with the E protein. The Western blot in Fig 1 shows a very light band for the Envelope protein, suggesting little to no interaction of prM to E protein pulled down with cholesterol. The authors should provide an explanation.

[Response] The Western blot in Fig. 1 represents the crosslinking efficiency of the cholesterol probe to proteins in proximity ($<3\text{\AA}$). To study protein-protein interaction one should have used a lysis buffer without/mild detergents (which we have done for the Co-IP in Fig. 7). For the lipid-protein IP we have used harsh conditions of detergents to remove protein-protein interactions to ensure accurate purification of proteins crosslinked to the lipid probe. For example: lysis occurred in 1% SDS, reconstituting pelleted proteins occurred in 4% SDS and washing of the beads was done 10x with 1% SDS. Given these conditions protein-protein interactions should be dissolved. We have modified our manuscript and have given a statement for the E band observed in Fig. 1.

Cryo-EM structures of ZIKV purified from mammalian cells show the presence of lipid molecules close to prM-E membrane helices. The authors have not discussed this aspect in the manuscript. A comparative analysis of the possible cholesterol binding sites described by the authors in this manuscript and the lipids present in the cryo-EM structure should be included in the discussion.

[Response] We now have included this aspect in our discussion.

Line 100: How does data from CARC3 mutant fit in this model?

[Response] Cholesterol biosynthesis occurs in the ER, and Leier and colleagues (2020, doi: <https://doi.org/10.1038/s41467-020-17433-9>) have shown that upon ZIKV infection cholesterol, apart from many other lipids, is dysregulated. According to our data we propose a model that the ER comprises micro domains, which are rich in cholesterol and that such microdomains accumulate prM and E proteins. If the interaction with cholesterol is reduced,

prM still inserts into the ER membrane, but is not “anchored” at a specific location, thus negatively affecting efficient virus assembly. The interaction between cholesterol and prM could be an initiation step for particle assembly. In this context Zhang and colleagues (2019, doi: 10.1128/mBio.02375-19) have shown that the nucleocapsid is recruited to the site of virus assembly via NS2A (which contains CRAC/CARC motifs in its TMS). If, however, prM is insufficiently placed at such micro domains, the NC is insufficiently incorporated. For this we propose that the cholesterol-prM interaction creates a platform, which is needed for proper virus assembly. In fact, such a model has been reported for influenza virus (Scheiffele et al. (1999), doi: <https://doi.org/10.1074/jbc.274.4.2038>) and BVDV (Callens et al. (2016), doi: <https://doi.org/10.1371/journal.ppat.1005476>).

Line 137: check the amino acid numbering: aa 38-44 of ZIKV.

[Response] We have adapted our manuscript with respect to the aa numbering and the CRAC/CARC motifs.

Lines 293-300: ZIKV spread and CARC2 with the cholesterol solubilizing drug:
After how many days/hr was the virus spread analyzed?

[Response] In this experiment we did not look at virus spread. Instead, we looked at the ability of the CARC2 mutant viruses to infect M β CD treated cells. Infection was assessed at 24 hours.

Line 328: The authors say, “CARC3 mutants were severely affected independent of cellular cholesterol levels”. Does this mean that CARC3 is not a bona fide cholesterol-binding site? Results show the CARC3 mutations and the resulting loss of C-E colocalization might result from a structural change to M protein rather than any effect of cholesterol binding. Text in the results and discussion are confusing regarding the roles of these two sites and cholesterol.

[Response] We cannot exclude that structural changes might arise upon mutagenesis of the M domain in the prM protein. However, our MD simulation data suggest that these changes are minor, if any. The full picture can only be obtained via x-ray crystallography. Since our antibodies binding to capsid and prM are both polyclonal rabbit antibodies, we could not perform co-staining of these two proteins. However, our data (Supplementary Fig. 12) shows that prM is co-occurring with E, thus E can be used as surrogate for prM positive sites. As mentioned above, a rescue experiment was not feasible.

Given these technical limitations, we have rephrased our manuscript and hope that the reading was improved.

Line 279: The statement that “The results showed that within the first 48 hours when no virus spread occurs, and only viral replication in transfected cells is monitored” is not an accurate statement based on the published growth kinetics of ZIKV.

[Response] The published growth kinetics are with respect to infection and an MOI exceeding 1 (Cortese et al., 2017, doi: 10.1016/j.celrep.2017.02.014). With respect to the reverse genetics system (Münster et al., 2018, doi: 10.3390/v10070368) VeroE6 cells were used for

characterization of the system. In fact, comparison between Huh7-Lunet and VeroE6 cells highlights that VeroE6 growth kinetics are faster (CPE occurs at 96 hours in VeroE6 cells, Huh7-Lunet show CPE at 120 hours post transfection). The transfection efficiency using electroporation is <5% (Huh7-Lunet), thus, even if new virus particles are released between 24 and 48 hours the changes in RNA levels/*Renilla* luciferase activity counts cannot be distinguished from the biological variance between replicates.

Fig 5b. IFA is not of good quality, which makes the data unreliable. The scale bar is 10 cm. How does this scale bar compare to that in Fig 4d, where the scale bar is 20 μ M?

[Response] The images in Fig 5b and Fig4d have been updated to have similar scale bars of 200 μ m.

Cell-to-cell spread of ZIKV can be demonstrated with figures of higher magnification which will also improve the quality and show cell boundaries and nuclei in better resolution. Both Figs 5b and 4d have no DAPI signal. The authors should improve both figures to make the data more reliable.

[Response] The micrographs showing virus spread have now been optimized. We would like to point out that the nuclei had been stained with DAPI which is represented in grey.

Fig 5f is not convincing enough to conclude that CARC2 has a role in virus entry. The data show the drug affects all three viruses similarly. Authors should provide quantitative data to prove their point. Why were CRAC3 mutants not included in this experiment?

[Response] We agree with Reviewer 2 that M β CD treatment affects all viruses. Li and colleagues have shown that M β CD treatment negatively affects ZIKV infection, and in our study, we make the claim that CARC2 mutants are more affected when cholesterol levels change. To specify the defect of CARC2 mutants we have performed fusion assays in which ZIKV WT and CARC2 mutants were labeled with R18. We found that the mutants prematurely fuse at the vicinity of the plasma membrane likely leading to improper/insufficient delivery of the viral genome into the cytoplasm. These data have now been added to the manuscript as a new panel in Fig. 5.

CARC3 mutants were not included in this experiment as these mutants' showed defects in virus assembly (Fig. 6), which limited our ability to produce virus stocks.

Minor points

Line 67, Line 77: Please provide references for ZIKV NC.

[Response] A reference has now been added. ([https://www.jbc.org/article/S0021-9258\(21\)00861-9/fulltext](https://www.jbc.org/article/S0021-9258(21)00861-9/fulltext))

Line 73: Is it clear whether precursor capsid is associated with LD as LDs are made of monolayers and precursor C has a TM domain?

[Response] The capsid TM is the signal sequence for prM which gets cleaved upon translocation of the prM protein into the lumen of the ER. Subsequently, the signal sequence is

cleaved releasing the mature capsid into the cytoplasm. For the interaction of capsid with LD, the signal sequence can remain uncleaved by NS2B/3 and thus as LDs are formed capsid could associate with LDs. This is hypothetical and not experimentally proven. Apart from that capsid could associate with LDs the same way as it interacts with the lipid bilayer in the virus particle – via its alpha helix 1, which is the more likely explanation.

Line 272: It is not clear how the authors can state that the binding assays show that E protein conformation is intact and not affected by the mutations in the M domain of prM.

[Response] If E conformation would have been altered, receptor binding/attachment of virions to the cell surface should have been negatively affected. However, we did not observe changes in virion binding between WT and the mutants as determined by quantifying RNA genome numbers bound to the cells. In addition, we now included neutralization assays to address the reviewer's question further. We found that WT and CARC2 mutants were equally neutralized, suggesting that the overall structure of E mediating the initial steps of virus infection and being targeted by the used neutralizing antibody remain intact. These data have now been added to the Supplementary Fig. 6.

Line 279: Is it true that ZIKV does not release virus in 48 hours? It will be useful to provide a standard growth kinetic analysis of WT and mutant viruses including 0-120 hours with 24 h intervals, as majority of the conclusions are based on virus release.

[Response] We have rephrased our statement. In Fig. 6 we can clearly see viruses being released at 48 h.p.t. whereas at 24 hours, virus amounts are below the detection limit. As mentioned above, ZIKV growth kinetics vary between the selected cell culture model, which has to be taken into account.

Line 300: Based on the mechanism of flavivirus entry, it is not clear how cholesterol binding of M protein affects virus entry as M is not exposed during entry, and access is entirely dependent on E protein.

[Response] It is true that virus entry is mainly known to be mediated by E-proteins and its fusion loop. There have been reports in which Semliki Forest virus was *in silico* fused with liposomes having different lipid composition (White & Helenius 1980, doi: 10.1073/pnas.77.6.3273). In this study it became clear that when cholesterol was removed fusion was fully abrogated. This result already suggested that apart from the insertion of the fusion loop and the folding back of E-proteins, one more step must occur to promote the fusion of the viral envelope with the endosomal membrane. We propose a model in which membrane fusion initiates, but at insufficient cholesterol concentration gets frozen in a hemifusion state. During the transition from hemifusion to fusion pore formation, the M protein might facilitate lipid mixing or specifically interacts with cholesterol to create a pore as it is reported for bacterial toxins. We have added these considerations in the discussion section.

Extended Data Figure 2A shows more virus is released from the mutants at early time points. It is important to show growth kinetic assays, as 120 hours seems too late to have cell viability. Does the specific infectivity change over time?

[Response] We agree that 120 h.p.t. is a very late time point for ZIKV growth kinetics. We did perform a cell viability assay to address the question whether WT and CARC2 mutants differ in their CPE. Our results show no significant differences between WT and mutants, arguing that the reduced virus titer is due to minimal, but steadily accumulating defects of infecting the bystander cells.

We compared the specific infectivity of WT and the CARC2-S mutant at 72 h.p.t. finding similar results with respect to a decrease for the CARC2-S mutant (Rebuttal Fig. 8).

Rebuttal Fig. 8. Specific infectivity was assessed at 72 h.p.t. for WT and CARC 2-S mutant. Huh7-Lunet cells were transfected with synZIKV WT or CARC 2-S and at 72 hours, supernatants were collected. Titers were determined by TCID50 assays. Extracellular RNA was estimated by probe based qRT-PCR. Specific infectivity (titers/viral genome copy numbers) are plotted from three different experiments.

Line 357: Since the virus release is not significantly reduced in the mutant virus, how is the reduction in virus particles calculated by EM analysis explained?

[Response] We apologize to the confusion of our nomenclature with respect to the mutants. We have optimized our manuscript and hope that the reading was made easier.

Line 357 refers to CARC3 mutants and these are clearly reduced in virus titer (Figure 6a).

Virus release is significantly reduced for mutants of the CARC3 motif (Fig. 6a), also virus particles made within the cell (Fig. 6c) were significantly lower when compared to WT. We used a reporter cell line which upon ZIKV infection depicts a green nucleus. This approach allowed us to specifically investigate ZIKV positive cells via EM. Here, we analyzed three consecutive sections (one section is 70 nm thick, total thickness analyzed: 210 nm) and estimated the numbers of virus particles within these cells. Furthermore, as confirmation we measured the particles and their size matched the one which was published (Cortese et al., 2018, doi: 10.1016/j.celrep.2017.02.014).

Please provide a catalog number for PAC-cholesterol purchased from Sigma Aldrich.

[Response] Photo-reactive Clickable trans-Sterol Probe (804657-5MG) has been added to the Supplementary Information.

[Response] To this end cholesterol was neither found in the existing cryo-EM structure analysis nor in lipidomic profiles of flaviviruses. Possible reasons for this are: e. Identification of lipids and lipid pockets in Spondweni virus (Renner et al. 2021, <https://doi.org/10.1038/s41467-021-21505-9>), dengue and West Nile virus (Hardy et al.. 2021, <https://doi.org/10.1038/s41467-021-22773-1>). These structural analyses were performed on virus particles derived from mosquito cells, which are lacking enzymes essential for de novo synthesis of cholesterol and lipidomic profiles of mosquito cells have not identified cholesterol, thus its detection in virions is not feasible. f. Apart from these two reports structural analysis was done on ZIKV (Sirohi et al., 2016, doi: 10.1126/science.aaf5316; Seevana et al., 2018, doi: 10.1016/j.str.2018.05.006), yet these reports were solely focusing on the structural organization of the particles and do not provide information towards lipids. g. In the study by DiNunno and colleagues (2020, doi: 10.1038/s41467-020-18747-4), which analyzed the lipid pockets found in ZIKV particles recovered from mammalian cells, the authors described a differential lipid pocket. This pocket arises in virions derived from mammalian cells but is absent in virions derived from mosquito cells. However, the authors do not make a claim of a specific lipid that is incorporated but instead state that it appears to be a crooked lipid.

REVIEWERS' COMMENTS

Reviewer #1 (Remarks to the Author):

The authors have adequately responded to all of my comments/suggestions in the revised ms. Accept.

Reviewer #2 (Remarks to the Author):

The authors have carefully considered and addressed the comments in the revised manuscript. The revised manuscript requires no changes.

Reviewer #3 (Remarks to the Author):

This is a revised manuscript from Ralf Bartenschlager and colleagues detailing the identification of functional cholesterol binding motifs within the structural protein prM of Zika virus. The original set of three reviews expressed interest in the work but had a significant number of queries. In an ~ 20 page response, the authors proceeded to address each of the questions/concerns that were raised. In some cases, new experiments that provided clarity were undertaken and in other cases the text was rewritten for clarity. After extensive review of the queries and responses, I think the authors have appropriately responded and have enhanced the impact and accuracy of the manuscript. This is an important finding and will add significantly to the flavivirus field as well as other enveloped viruses.